# Neuron-based explanations of neural networks sacrifice completeness and interpretability

**Nolan Dey**                                                                                   *nolan@cerebras.net*
*Cerebras Systems, University of Waterloo, Vector Institute*

**Eric Taylor**                                                                        *eric.taylor@vectorinstitute.ai*
*Borealis AI, Vector Institute*

**Alexander Wong**                                                                      *alexander.wong@uwaterloo.ca*
*University of Waterloo, Apple*

**Bryan Tripp**                                                                                 *bptripp@uwaterloo.ca*
*University of Waterloo*

**Graham W. Taylor**                                                                          *gwtaylor@uoguelph.ca*
*University of Guelph, Vector Institute*

**Reviewed on OpenReview:** *https://openreview.net/forum?id=UWNa9Pv6qA*

## Abstract

High quality explanations of neural networks (NNs) should exhibit two key properties. *Completeness* ensures that they accurately reflect a network's function and *interpretability* makes them understandable to humans. Many existing methods provide explanations of individual neurons within a network. In this work we provide evidence that for AlexNet pretrained on ImageNet, neuron-based explanation methods sacrifice both completeness and interpretability compared to activation principal components. Neurons are a poor basis for AlexNet embeddings because they don't account for the distributed nature of these representations. By examining two quantitative measures of completeness and conducting a user study to measure interpretability, we show the most important principal components provide more complete and interpretable explanations than the most important neurons. Much of the activation variance may be explained by examining relatively few high-variance PCs, as opposed to studying every neuron. These principal components also strongly affect network function, and are significantly more interpretable than neurons. Our findings suggest that explanation methods for networks like AlexNet should avoid using neurons as a basis for embeddings and instead choose a basis, such as principal components, which accounts for the high dimensional and distributed nature of a network's internal representations. Interactive demo and code available at `https://ndey96.github.io/neuron-explanations-sacrifice`.

## 1 Introduction

Large neural networks (NNs) are increasingly consequential. For example, widespread use of large language models brings a wide range of societal risks (Weidinger et al., 2022; Shevlane et al., 2023), and NNs are increasingly used in high-stakes application areas such as healthcare (Rajpurkar et al., 2022), including in many FDA-approved devices [1]. Despite their increasing importance, we still do not have satisfactory methods to explain how NNs arrive at their outputs. Many sensitive applications would benefit from reliable

---

[1] `https://www.fda.gov/medical-devices/software-medical-device-samd/artificial-intelligence-and-machine-learning-aiml-enabled-medical-devices`

explanations of neural networks. Developing improved methods of explaining NNs can also advance our understanding of neural representations and lead to improved training techniques.

At the heart of the explainability challenge is the tradeoff between explanation completeness and explanation interpretability (Gilpin et al., 2018). Completeness describes how accurately and thoroughly an explanation conveys a model's function while interpretability describes how easy it is for a human to understand an explanation. The most complete explanation would be to simply display the equation for a layer's forward pass. However, this explanation has poor interpretability. At the opposite extreme, many popular DNN explanation methods make choices that increase interpretability at the expense of completeness. For example, saliency map and feature visualization explanations (Simonyan et al., 2013; Olah et al., 2017) are highly interpretable but offer unreliable explanations (Kindermans et al., 2017; Geirhos et al., 2023).

The problem of understanding a NN as a whole can be decomposed into understanding the non-linear transformation applied by each layer. Each NN layer applies a non-linear transformation to input activation space to produce an output activation space. Understanding any such transformation means understanding how the output space is produced relative to input space. Unfortunately, humans cannot directly understand the meaning of a point in the internal representations used by the network. Instead humans are capable of interpreting a point in a *network's* input space (e.g. an image, sentence, or row of tabular data). This motivates the use of visualization methods that map hidden activations back to a network's input space. Some popular visualization methods are feature visualization (Olah et al., 2017), nearest neighbors search (Karpathy, 2014), and more complex generative modeling approaches (Rombach et al., 2020; Nguyen et al., 2016b). Using an activation visualization technique further narrows the problem of explaining the $l$th layer's transformation into understanding how the $l$th layer's transformation maps back to the network's input space.

To solve this problem we need to select points to visualize and interpret that are representative of activation space. A practical way forward is to choose a basis and visualize points along each basis vector separately. However, the problem remains difficult because:

- The curse of dimensionality - hidden activations commonly have thousands of dimensions.

- Like biological ones, artificial neural networks have distributed representations; multiple neurons fire together to represent concepts (Barrett et al., 2019), (Fong & Vedaldi, 2018), (Gallego et al., 2017), (Härkönen et al., 2020), (Leavitt & Morcos, 2020), (Olah et al., 2018), (Rombach et al., 2020).

- Neurons can be poly-semantic - a single neuron can represent multiple concepts (Olah et al., 2020b).

- Human attention is a limited resource - the number visualizations that can be interpreted is limited.

Basis selection must take each of these difficulties into account to support relatively complete and interpretable explanations.

The most popular choice of basis is the standard basis which corresponds to focusing on understanding each individual neuron in isolation. The neuron basis disregards the distributed nature of NN representations, detracting from explanation completeness. Furthermore, studying each neuron in a layer requires an impractical amount of human attention. Explanation methods that focus on individual neurons (e.g. Bau et al., 2017; 2020; Mahendran & Vedaldi, 2015; Mu & Andreas, 2020; Nguyen et al., 2016a;b; Olah et al., 2017; 2018; Bills et al., 2023; Schwettmann et al., 2023; Dai et al., 2022) typically only provide interpretable explanations for neurons that are highly selective of some concept. However, only a small fraction of individual neurons are highly concept-selective (Bau et al., 2017), (Olah et al., 2017). Furthermore, highly concept-selective neurons can be ablated without removing a model's ability to recognize that concept, meaning neurons that are not highly concept-selective still play an important role in network function (Barrett et al., 2019), (Donnelly & Roegiest, 2019), (Kanda et al., 2020), (Leavitt & Morcos, 2021), (Morcos et al., 2018). Finally due to the polysemantic nature of neurons, one would need to understand each concept that a neuron is involved in representing, requiring an even larger expenditure of human attention.

Instead, a powerful alternative is to study a basis obtained through a dimensionality reduction technique. This can be done by sampling hidden activations produced for each example in a large-scale dataset and

fitting a dimensionality reduction model to obtain the basis (e.g. Alammar, 2020; Karpathy, 2014; Carter et al., 2019). This general approach accounts for the distributed nature of NN representations, provides a greatly reduced set of basis vectors to analyze, and can take into account the polysemantic nature of neurons.

In this study we provide evidence illustrating the inferiority of the neuron basis compared to the principal component analysis (PCA) basis, a simple baseline. Our method can be applied to feed-forward networks as well as CNNs and we expect that it can be adapted to other architectures including Vision Transformers Dosovitskiy et al. (2021). While our methods are widely applicable, we focus this paper on studying AlexNet (Krizhevsky, 2014) pretrained on ImageNet (Deng et al., 2009; Paszke et al., 2019) because it is studied in several related explainability works (Bau et al., 2017), (Fong & Vedaldi, 2018), (Mu & Andreas, 2020), (Zhou et al., 2018), (Rajpal et al., 2023) and its limited depth makes it feasible for us to study each layer in detail in a user study.

To compare the explanations produced using the neuron and PCA bases, we propose two quantitative measures of completeness and conduct a user study to measure explanation interpretability. Firstly, we show that small numbers of PCs account for much of the activation variance in each layer (Section 4.1.1). Also, through ablation experiments we show that PCs with large eigenvalues are generally more important for AlexNet's (Krizhevsky, 2014) performance than the most important neurons, and that PCs with small eigenvalues are generally less important for AlexNet's performance than the least important neurons (Section 4.1.2). Through a user study we also show that PCs with large eigenvalues tend to be interpretable and PCs with small eigenvalues are generally uninterpretable. Moreover, the most important PCs are more interpretable than the most important neurons (Section 4.2). In summary, our results show the most important PCs form a more complete and interpretable basis than the most important neurons.

## 2 Related work

Several prior works have also examined the validity of neuron-based explanations. Morcos et al. (2018) perform cumulative neuron ablations that show highly class-selective units may be harmful to network performance. Zhou et al. (2018) follow this up by conducting neuron ablations to show that ablating highly class-selective individual units hurts the accuracy of specific classes more than overall accuracy, supporting the view that it is meaningful to study these neurons. Leavitt & Morcos (2021) continued this line of inquiry by showing that regularizing against learning class selective neurons can improve accuracy. Radford et al. (2017) demonstrate the existence of a "sentiment neuron" which is highly selective of text sentiment. However, Donnelly & Roegiest (2019) find that removal of the "sentiment neuron" only marginally affects sentiment classification accuracy and may even improve it.

Morcos et al. (2018); Leavitt & Morcos (2021); Donnelly & Roegiest (2019); Barrett et al. (2019) caution that methods for understanding neural networks based on analyzing highly selective single units, or finding optimal inputs for single units (e.g. Olah et al., 2017) may be misleading.

Another promising line of inquiry is Concept Activation Vectors (CAVs) (Kim et al., 2018), which are activation vectors that align with pre-defined human-interpretable concepts. CAVs provide interpretable explanations because they take into account the distributed nature of representations. However, CAVs can lack completeness because a user must predefine concepts to explain (similar to Network Dissection (Bau et al., 2017), making it unlikely to encompass the full set of concepts represented by a NN. Ghorbani et al. (2019) introduce automated concept-based explanations (ACE) to automate the process of defining concepts through an activation clustering algorithm. Both TCAV and AutoTCAV find directions that align with concepts but do not take into account overall explanation completeness. To remedy this, Yeh et al. (2020) propose ConceptSHAP to measure completeness and find concepts to maximize this metric. Notably they find the completeness of concepts found via PCA is comparable or better than ACE but their method still outperforms PCA completeness. They show PCA maximizes the $\ell_2$ surrogate of their completeness score but PCs lack human interpretability. In contrast, we find PCs can be highly interpretable if they are fit to activations prior to non-linearities (see Section 3.1).

Both biological and artificial neural networks have distributed representations; multiple neurons fire together to represent concepts (Barrett et al., 2019; Fong & Vedaldi, 2018; Gallego et al., 2017; Härkönen et al., 2020;

Leavitt & Morcos, 2020; Olah et al., 2018; Rombach et al., 2020). A basis vector must have the ability to take into account the firing of multiple neurons at once. Barrett et al. (2019) highlight that population level analysis is useful for BNNs and ANNs and suggest dimensionality reduction could be a way forward.

The study of individual neuron responses is an approach inherited from neuroscience, where stimulus tuning of individual neurons has been studied since the 1960s (Hubel & Wiesel, 1962; Dayan & Abbott, 2005), and the reverse correlation method has been used to determine stimuli that optimally drive sensory neurons since the 1970s (De Boer & De Jongh, 1978; Dayan & Abbott, 2005). A focus on individual neurons was unavoidable at first because multi-neuron recordings were technically difficult. However, the importance of understanding neural activity at the population level has been recognized for decades (Georgopoulos et al., 1986; Zemel et al., 1998; Schneidman et al., 2003). Meanwhile, advancing methodology has led to a doubling in the number of neurons that can be simultaneously recorded about every seven years (Stevenson & Kording, 2011), driving a shift away from single-neuron analyses to population analyses (Saxena & Cunningham, 2019; Ebitz & Hayden, 2021; Urai et al., 2022), particularly in the last fifteen years. Linear dimensionality reduction is valuable for studying the activity of large populations. For example, this approach has shown that population activity in the primate motor cortex during reaching is largely explained by low-dimensional rotational dynamics (Shenoy et al., 2013), and that related dynamics are important in perceptual decision making in the prefrontal cortex (Aoi et al., 2020). Activity in mouse primary visual cortex is correlated so that the $n^{th}$ principal component variance decays as $1/n$ (Stringer et al., 2019), or more quickly for smaller components (Pospisil & Pillow, 2024).

## 3 Methodology

As described in Section 1, the problem of explaining a NN can be decomposed into understanding the nonlinear transformation applied by each layer in terms of the NN's input space. To facilitate this for a particular layer, we sample activations, fit a basis for activation space, and visualize points along each basis vector. This is summarized in Figure 1. Our methodology is computationally inexpensive and simple to apply to any feed-forward or convolutional layers.

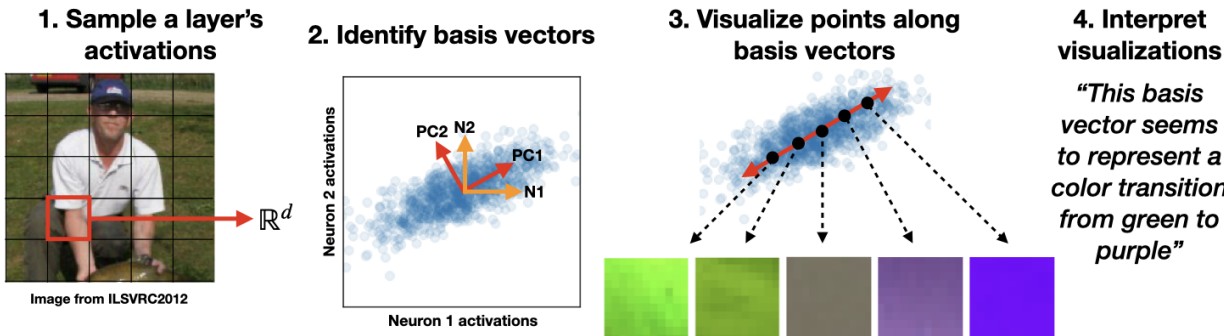

Figure 1: Overview of our methodology. We sample $\mathbb{R}^d$ activations from a layer (1), then identify basis vectors of the layer's activation space (e.g. neurons or PCs) (2). Finally we visualize points along each basis vector (3) and interpret the visualizations (4).

### 3.1 Sampling a layer's activations

We performed PCA on a large representative sample of activations obtained by forward propagating every image in the ImageNet training set through a DNN up to a specified layer. We used an activation matrix $\mathbf{A} \in \mathbb{R}^{n \times d}$ where $n$ is the number of training examples, and $d$ is either the number of convolutional channels or the number of fully-connected neurons, depending on the layer type. For fully-connected layers, each row of the matrix consisted of the full $\mathbb{R}^d$ response to the corresponding image, where $d$ is the number of neurons. For convolutional layers, since the same operation is applied to each spatial position, we sampled a single spatial position from each channel. Sampling activations from the same spatial position for every

image could bias our sampling towards input features typically found at that position. To avoid this, we chose random spatial positions for each image, avoiding boundary positions. We adopted this method of sampling activations from (Carter et al., 2019). We also found sampling activations before the non-linearity allowed for much better PCA fits and interpretable visualizations.

## 3.2 Identifying basis vectors

Activation space is high dimensional, often with thousands of dimensions. The neuron basis, defined by axis-aligned unit vectors corresponding to each neuron, is the naïve choice of basis for activation space. Using dimensionality reduction we can find bases that capture more information with fewer basis vectors. To apply dimensionality reduction to a large scale $\mathbf{A}$, we require an invertible method that has computationally inexpensive fit, transform, and inverse transform operations. For the purpose of this study, we believe studying the PCA basis is sufficient to illustrate the shortcomings of the neuron basis. Alternative dimensionality reduction methods such as ICA (Hyvärinen & Oja, 2000), NMF (Cichocki & Phan, 2009), LLE (Roweis & Saul, 2000), kPCA (Schölkopf et al., 1997), VAE (Kingma & Welling, 2014), t-SNE (van der Maaten & Hinton, 2008), and UMAP (McInnes et al., 2018) require the number of components to be known beforehand, are too computationally expensive to apply to $\mathbf{A}$, have many sensitive hyperparameters, and/or do not possess an analytical inverse.

Using scikit-learn (Pedregosa et al., 2011), we apply PCA to $\mathbf{A}$ to obtain transformed activations $\mathbf{A}' \in \mathbb{R}^{n \times p}$ where $p$ is the number of principal components (PCs). PCA finds orthogonal basis vectors for $\mathbb{R}^d$ activation space, ordered by explained variance ($\mathrm{Var}(\mathbf{A}'_i)$ for the $i$th basis vector). If activations have some covariance structure, then more activation variance can be explained with relatively few PCs. Significant activation covariance also implies neuron interactions are responsible for much of the activation variance.

## 3.3 Visualizing points along basis vectors

To gain insight into what each basis vector represents, we visualize $m$ points along each basis vector independently of any other basis vectors. By projecting $\mathbb{R}^d$ activations onto $p$ basis vectors, we obtain a transformed activation space $\mathbb{R}^p$. We uniformly sample $m$ points between the observed minimum and maximum values along each basis vector to obtain an $(m \times p)$ matrix of sampled points in transformed activation space. Then, we apply an inverse transform to obtain an $(m \times d)$ matrix of sampled points in untransformed activation space. See Appendix C for a comparison of sampling methods.

Karpathy (2014) visualized a point in activation space $\mathbf{a}_{\text{target}} \in \mathbb{R}^d$ by displaying the receptive-field sized image patch whose activations have with the lowest $\ell_2$ distance to the point $\mathbf{a}_{\text{target}}$. In this work we extend Karpathy's method by displaying the $k$ nearest receptive-field sized image patches that have the lowest $\ell_2$ distance to $\mathbf{a}_{\text{target}}$ when forward propagated. We perform this search over the $N$ receptive field-sized image patches in $\mathbf{A}$ using the Faiss library (Douze et al., 2024). It is possible that adjacent points along a basis share the same nearest neighbor.

Figures 2, 3, and 4 show examples of AlexNet PC visualizations, where left to right moves from negative to positive values along each PC. For early network layers where the size of the receptive field is small, the visualizations are quite interpretable (Figure 2). For deeper layers with larger receptive fields, it becomes more difficult to interpret what features the visualizations have in common because there is more area for distractors to be present. To help increase the signal to noise, we mask away the regions of the visualizations that do not strongly affect the proximity of the representation to the point $\mathbf{a}$ in representation space that we are visualizing (Figure 3). See Appendix B for more details on the masking procedure.

We also experimented with optimized synthetic "feature visualizations" following Olah et al. (2017). We found synthetic visualizations become uninterpretable after the first layer, corroborating the findings of Borowski et al. (2020). Borowski et al. (2020) conducted a user study and found natural image explanations to be more interpretable than optimized feature visualizations (Olah et al., 2017).

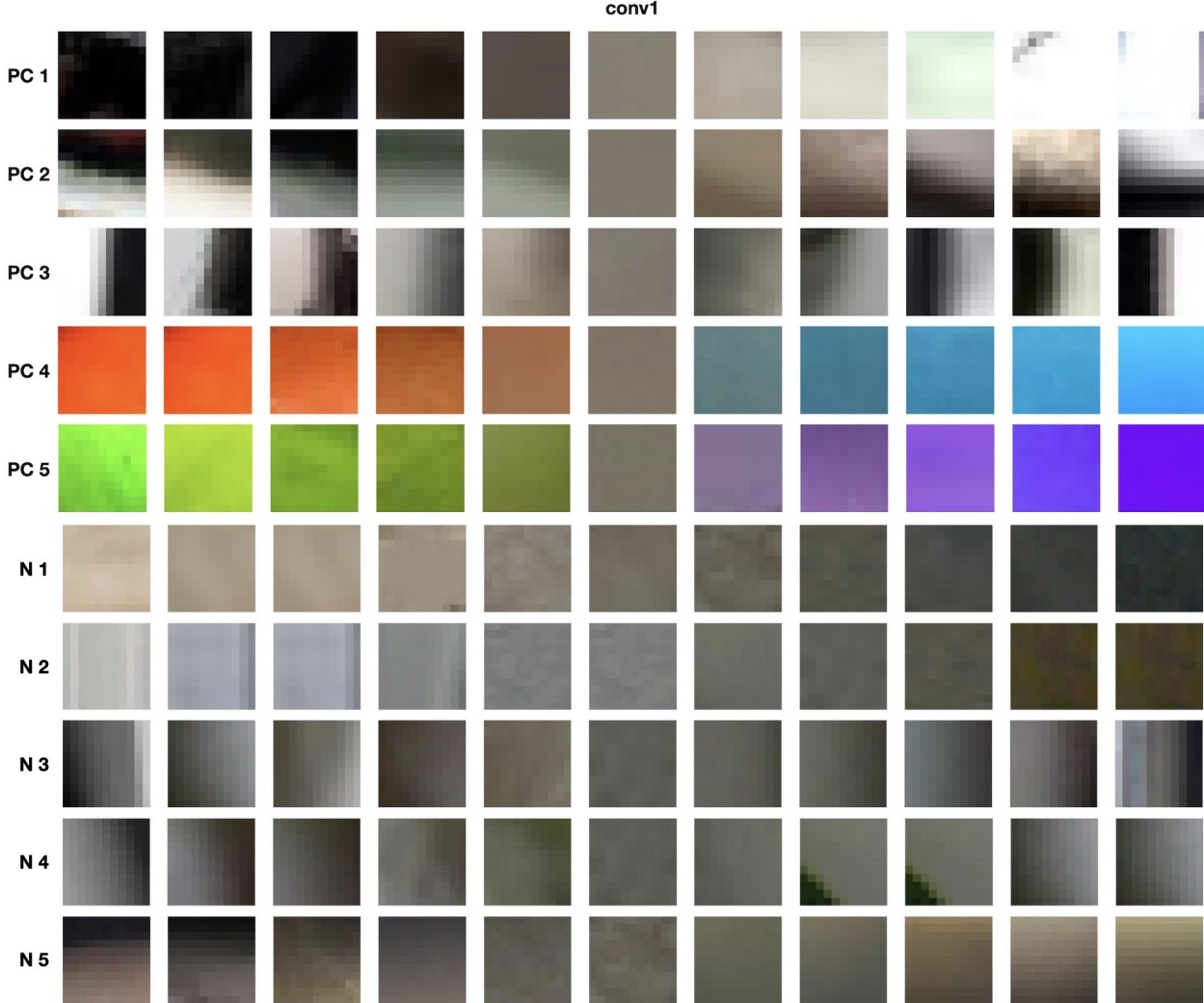

Figure 2: Visualizations of points along the 5 highest variance PCs and 5 highest variance neurons for AlexNet's `conv1` layer.

### 3.4 Interpreting visualizations

Finally we must invest the precious resource of human attention to interpret the resulting visualizations. When visualizing activation space points we expect to see semantically meaningful changes as we move along a useful basis vector. For example, Figures 2 and 3 show visualizations from the first 2 layers of AlexNet. Moving from negative regions along the PC to positive regions (left to right), the visualizations change in some meaningful way. In these layers we see the angle of edges may change from horizontal to vertical, the color may change from orange to blue, patterns may change from vertical/horizontal to diagonal, etc. This can help give an idea of what features the DNN has learned to extract from the input to make classification decisions. To quantify the interpretability of each dimension, we report the results of a systematic study with human participants (see Sec. 4.2). Additionally, Appendix G contains visualizations of the 5 highest variance PCs and neurons for each AlexNet layer. The PCs for `conv1` capture features such as brightness, Gabor filter phase, color contrasts, and color center surrounds. The PCs for `conv2` capture features such as texture frequency, line orientation, and color. These results are consistent with the findings of a study on early convolutional layers (Olah et al., 2020a). The PCs for `conv3` - `conv5` capture more high level features such as textures and objects. Finally, the PCs of `fc1` - `fc3` seem to separate classes. It is encouraging that

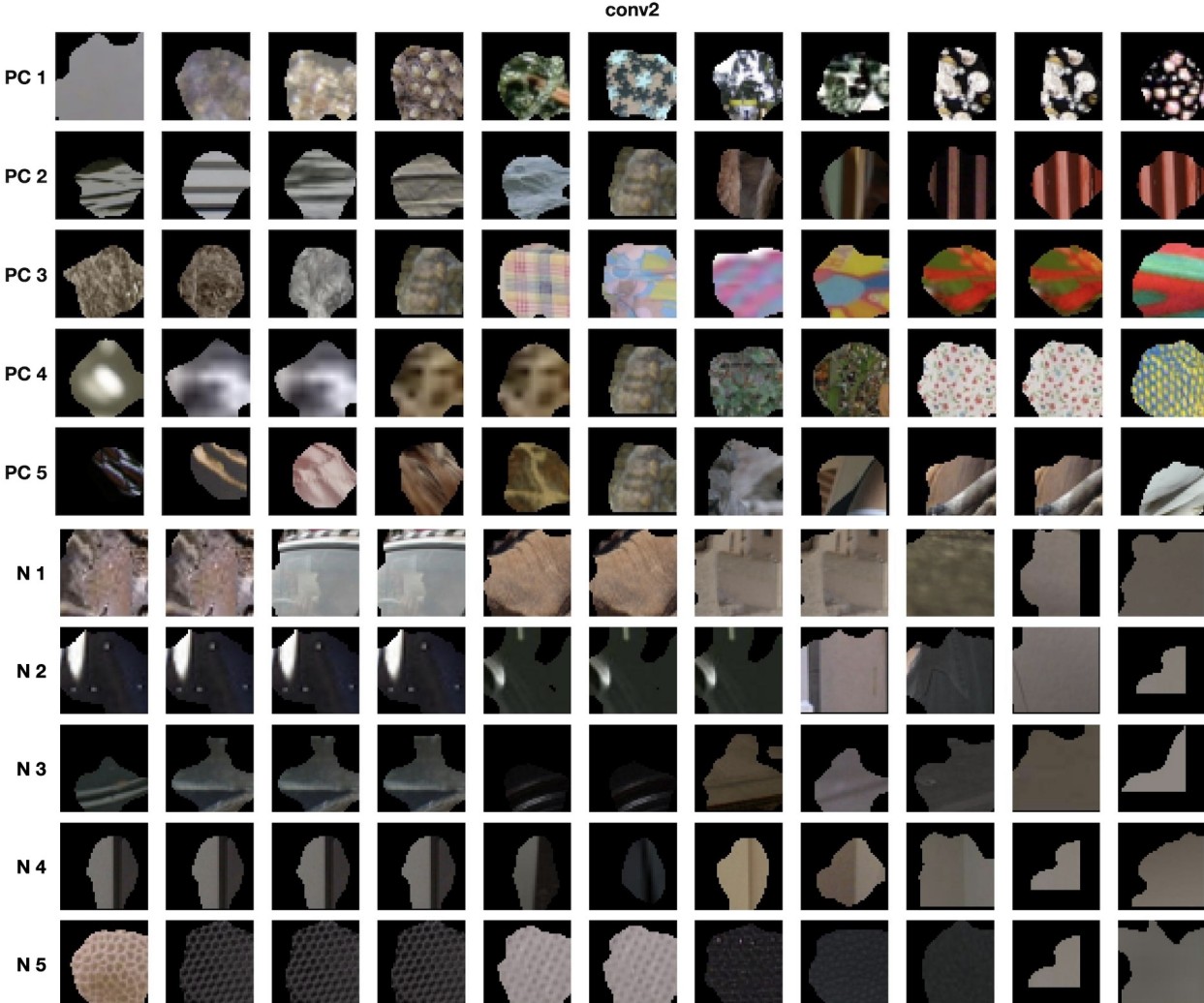

Figure 3: Visualizations of points along the 6 highest variance PCs and 5 highest variance neurons for AlexNet's `conv2` layer.

our method aligns with the general understanding of feature hierarchies in CNNs. In contrast, the neuron visualizations from `conv3` - `fc3` appear semantically meaningless (Appendix G).

Since it is difficult to exhaustively present these visualizations in a paper, we have also created an **interactive demo** (`https://ndey96.github.io/neuron-explanations-sacrifice`) including:

- PC and neuron visualizations for every layer of pre-trained AlexNet (Krizhevsky, 2014).

- PC visualizations for every residual block of pre-trained ResNet-18 (He et al., 2016).

- PC visualizations for every residual block of pre-trained ResNet-50 (He et al., 2016).

- PC visualizations for 10 layers of pre-trained ViT-B/16 (Dosovitskiy et al., 2021).

## 4 Experiments

Throughout this section we focus on comparing the PCA basis to the neuron basis. This comparison is important because the neuron basis is used in many influential explainable artificial intelligence (XAI) works

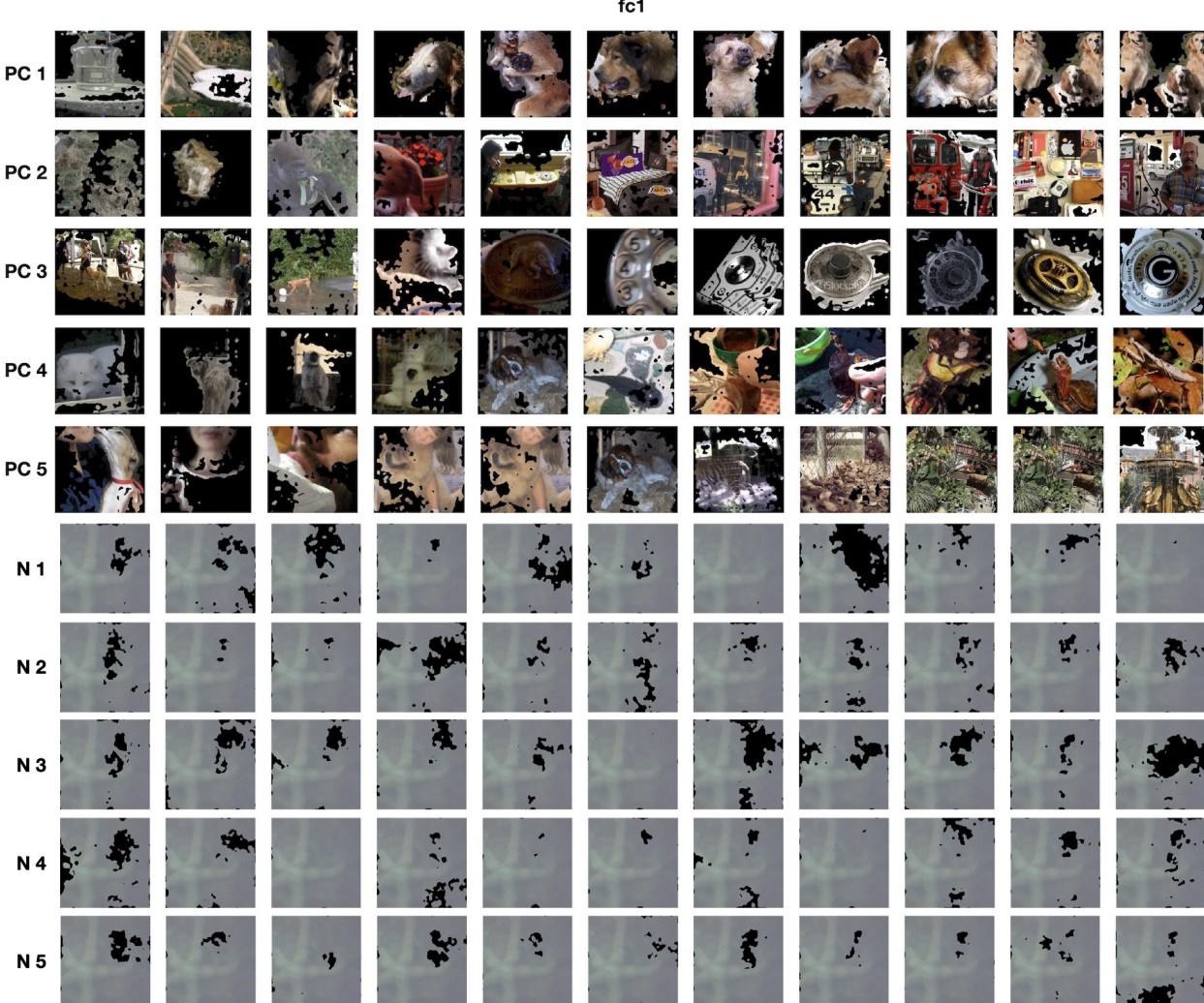

Figure 4: Visualizations of points along the 5 highest variance PCs and 5 highest variance neurons for AlexNet's `fc1` layer.

that attempt to study layer representations (Bau et al., 2017), (Bau et al., 2020) (Mahendran & Vedaldi, 2015), (Mu & Andreas, 2020), (Nguyen et al., 2016a), (Nguyen et al., 2016b), (Olah et al., 2017), (Olah et al., 2018), (Bills et al., 2023). In this work we study PyTorch's (Paszke et al., 2019) AlexNet (Krizhevsky, 2014) pretrained on ImageNet (Deng et al., 2009) because it is studied in several related works (Bau et al., 2017), (Fong & Vedaldi, 2018), (Mu & Andreas, 2020), (Zhou et al., 2018) and its depth makes it feasible to study each layer in more detail. AlexNet has five convolutional layers (`conv1` - `conv5`) followed by three fully-connected layers (`fc1` - `fc3`). To compare the neuron and PC bases, we use two quantitative measures of explanation completeness and conduct a user study to compare explanation interpretability.

## 4.1 Quantifying explanation completeness

Explanation completeness is an abstract concept that could be measured in a variety of ways. We use two complementary measures of subspace completeness below.

### 4.1.1 Activation covariance structure.

One measure of completeness is the fraction of activation variance explained by a set of basis vectors. In this context, explained variance refers to the amount of activation variance along a basis vector. In Figure 5, we plot the cumulative explained variance ratio of top-$k$ basis vectors which we define as:

$$\text{Cumulative sum of explained variance ratio of top-}k\text{ basis vectors} = \frac{\sum_{i=1}^{k} \text{Var}(\mathbf{A}_i')}{\sum_{i=1}^{n} \text{Var}(\mathbf{A}_i')} \quad (1)$$

where $\mathbf{A}_i' \in \mathbb{R}^p$. When measuring the explained variance of neurons: $\mathbf{A}' = \mathbf{A}$ (i.e. the activations). When measuring the explained variance of principal components, one could equivalently calculate

$$\text{Cumulative sum of explained variance ratio of top-}k\text{ basis vectors} = \frac{\sum_{i=1}^{k} \lambda_i}{\sum_{i=1}^{n} \lambda_i} \quad (2)$$

where $\lambda_i$ is the $i$th eigenvalue of $\mathbf{A}^T\mathbf{A}^2$. Figure 5 shows a comparison between the explained variance of the PCA basis and the neuron basis for AlexNet activations. Much of the activation variance is concentrated in the most important PCs whereas explained variance is far less concentrated in the neuron basis. For example, to explain 80% of the activation variance for `fc1`, one could either study the first 42 PCs, or the 2782 highest variance neurons. Similar trends are observed in every layer, showing PCA provides a more efficient basis for activation space. Notably, `conv4` and `conv5` require a similar number of PCs and neurons to explain 99% of the activation variance, implying these activations have a higher rank structure and PCA is a less suitable decomposition method compared to other layers. This stands in contrast to `fc1` and `fc2` which appear strikingly low rank.

The success of the PCA is due to significant covariance between neurons in each layer; Without inter-neuron covariance, the PCA basis would be roughly as efficient as the standard neuron basis for explaining activation variance. This measure naturally favors PCA, but the size of the difference is not known in advance. The dramatic difference seen here suggests that it is wasteful to omit dimensionality reduction when explaining neural networks.

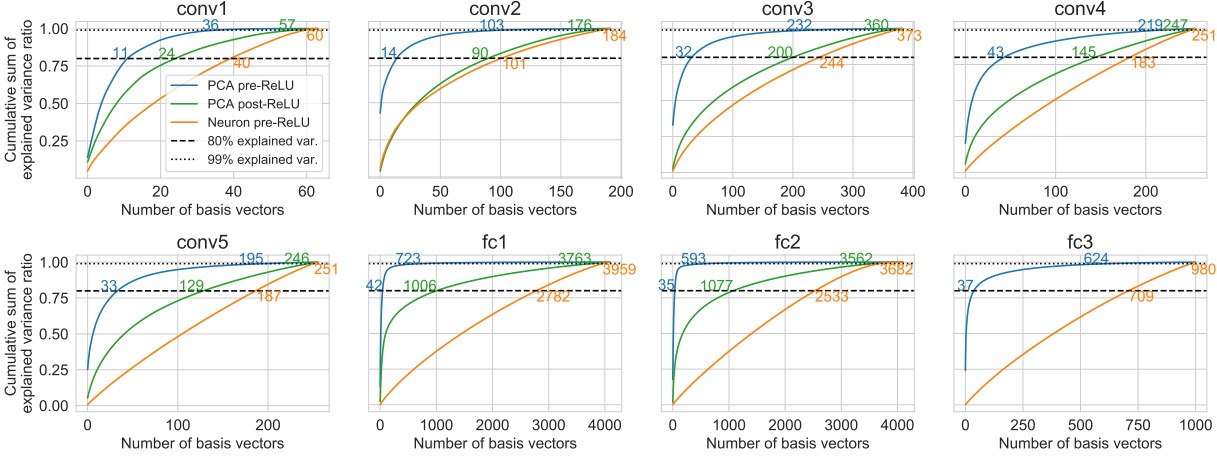

Figure 5: Cumulative sum of explained variance ratio for each AlexNet layer plotted against the number of basis vectors being used. Both PCs and neurons are ordered by descending variance. The number of basis vectors required to explain 80% and 99% variance is annotated.

### 4.1.2 Activation ablation.

Another quantitative measure of completeness is the amount of generalization ability that can be attributed to a set of basis vectors (Amjad et al., 2021; Morcos et al., 2018; Zhou et al., 2018; Raghu et al., 2017).

---

[2]As done in scikit-learn (Pedregosa et al., 2011)

To quantitatively test the importance of each basis, we cumulatively ablate basis vectors and observe how much accuracy degrades. Specifically, we cumulatively ablate both neurons and PCs, in both descending and ascending order of activation variance, then measure the effect on AlexNet's validation accuracy.

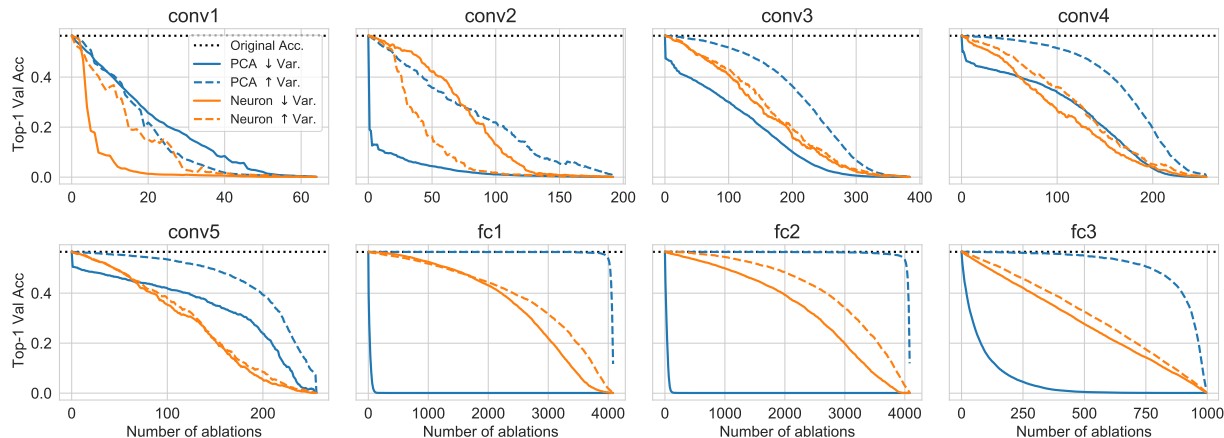

Figure 6: For each AlexNet layer, we ablate basis vectors in activation space and measure the effect on ImageNet top-1 validation accuracy. Both PCs and neurons are ordered by their explained variance. Reverse order corresponds to ascending explained variance.

Figure 6 shows that AlexNet accuracy degrades fastest when ablating the highest variance PCs, with the exception of `conv1`. Ablating high variance neurons in `conv1` is highly damaging because neurons in the first layer of CNNs learn fundamental vision features like edge detection and color contrasting (Bergstra et al., 2011), (Olah et al., 2020a). The lack of dropout in `conv1` means redundancy is not encouraged so ablating an important neuron destroys a vital feature for classification for the rest of the model depth. In contrast, since each PC is a linear combination of neurons, ablating a PC is unlikely to entirely destroy an important neuron's activations. The fully-connected layers `fc1,fc2,fc3` have highly redundant representations where a tiny fraction of PCs is responsible for performance, perhaps due to dropout and extreme width. This is evidence that the most important PCs offer a more complete explanation of activation space than the most important neurons, and the least important PCs contribute the least towards explanation completeness. As Figure 5 showed, `conv4` and `conv5` activations have heavier tailed eigenspectra and PCA may be a less suitable decomposition for these layers. We hypothesize this explains why ablating the ∼50 highest variance PCs degrades accuracy faster than neurons, but after that point it is more damaging to ablate neurons. Zhou et al. (2018) found that ablating random basis vectors from AlexNet `conv5` had less of an effect on accuracy than ablating individual neurons. We show that ablating the most important PCs has a greater effect on accuracy than individual neurons, and thus also has a greater effect than ablating random directions for `conv5`.

## 4.2 Quantifying explanation interpretability

Following best practices (Leavitt & Morcos, 2020; Gilpin et al., 2018), we measure interpretability via a user study to validate that humans can indeed interpret coherent stimuli along the visualized basis vectors.

We presented users with two visualizations of each PC/neuron (see Appendix F for screenshots). One visualization was randomly shuffled while the other was in the correct order. Participants were instructed to select the visualization that displayed a coherent transition from left to right. If participants cannot accurately determine which one is random, then they cannot interpret the stimulus dimension; the continuity of the visualization is its defining feature.

We tested the following hypotheses: (1) neuron-based visualizations are less interpretable than PCA visualizations, and (2) visualizations from shallow layers should be more interpretable than ones from deeper layers.

For both the PCA and neuron basis, we tested the 16 basis vectors with the largest variance from each of AlexNet's 8 layers. Stimuli were presented in random order. The position (top/bottom) of the original and scrambled stimuli was randomized. A visualization consisted of three rows of natural image snippets from the three nearest neighbours in activation space to points along each basis. Synthetic visualizations were not included due to screen space constraints and limited interpretability. The scrambled versions contained identical snippets in pseudo-random order. 22 user study participants were recruited. See Appendix F for more details.

Figure 7 summarizes the results of the user study: PC visualizations were, on average, more interpretable than Neuron visualizations for each layer in AlexNet with the most pronounced differences seen in layers `conv2`, `fc1`, and `fc2`.

To test the first hypothesis, we ran a one-way ANOVA with basis method (PCA vs Neuron) as a within-subjects factor. Results showed a significant advantage for perception of continuity for the PCA basis visualizations: $F(1,21) = 34.50$, $p < .001$, confirming our first hypothesis.

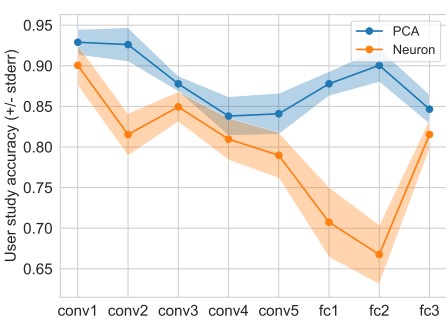

To assess the relative interpretability of visualizations for bases drawn from different layers, we ran separate post-hoc factorial ANOVAs for `conv` and `fc` layers as they serve qualitatively different functions. Among fully connected layers, there was an advantage for PCA over Neuron basis interpretability, $F(1,21)$ = 32.91, $p < .001$; an effect of layer depth that is difficult to interpret, $F(2,42) = 4.01$, $p < .05$; and an interaction between basis method and layer, $F(2,42) = 8.95$, $p < .001$, such that the difference appears pronounced in `fc1` and `fc2`, but less so in `fc3`. Among convolutional layers, there was an advantage

Figure 7: User study accuracy for each AlexNet layer. Shaded regions indicate the standard error across 22 study participants.

for PCA over Neuron basis interpretability, $F(1,21) = 9.32$, $p < .001$; an advantage for interpretability of visualizations for bases from shallower versus deeper layers, $F(4,84) = 10.66$, $p < .001$; and no significant interaction.

## 5    Limitations

**We mainly study AlexNet-ImageNet.** We focused on AlexNet pretrained on ImageNet because it is one of the most studied networks in the explainability literature (Bau et al., 2017), (Fong & Vedaldi, 2018), (Mu & Andreas, 2020), (Zhou et al., 2018). However, we did not compare the explanation completeness and interpretability of neuron and PCA based explanations for other networks or datasets. This potentially limits the generality of our claims regarding neuron-based explanations of neural networks beyond AlexNet. In Appendix A we include cumulative sum of explained variance plots for ImageNet-pretrained ResNet-18, ResNet-50, and ViT-B/16 (Figure 8, 9, 10). Similar to AlexNet, the activation variance is much more concentrated in the high variance PCs compared to the high variance neurons, suggesting our findings may generalize to these networks. In our online demo, we include visualizations of the top 16 PCs for several layers in ResNet-18, ResNet-50, and ViT-B/16.

**There are cases where neuron-based explanations might be useful.** In principle, a neuron basis could be more interpretable than a top PC basis. Such an example could be constructed with an auxiliary loss that explicitly aligns a neuron with a clear concept. However, standard network training does not particularly encourage this, and the probability of this happening on its own just seems to be low. Neuron-based explanations should be most useful in the first and final layer of a network. Given their linear nature, it is well known that the convolutional filter weights in the first layer can be directly interpreted as Gabor filters, color detectors, edge detectors, etc. Our results in Figure 6 show ablating the highest variance neurons in `conv1` is more damaging than ablating the highest variance PCs. In the final layer, the weights act as a classification head so each neuron has an interpretable meaning as it is associated with a particular class. However, our results in Figures 5, 6, and 7 show that the PC bases were more complete and interpretable than the neuron bases for the `fc3` layer of AlexNet. Finally, the activation ablation experiments suggest one

could prune the lowest variance PCs or neurons as a form of model compression. While pruning low variance PCs would better preserve accuracy in most layers, it is much easier to achieve hardware acceleration from pruning low variance neurons instead.

**We define a low threshold for interpretability.** The study in Section 4.2 tests for a low threshold of interpretability because participants do not necessarily need to understand the transition they see, even if they correctly recognize it as coherent. This means that poor performance on this task means that users truly cannot make sense of the visualization. We chose this test design because it was the highest threshold test we could conceive that had a well-defined ground truth. While our user study showed PC explanations were clearly more interpretable than neuron explanations, the insight it can give into deep network function is still rather limited.

**We only study the forward pass, not the backward pass.** Understanding the backward pass of DNNs can also provide insight into the function of each layer. Instead of sampling forward activations $a \in \mathbb{R}^d$, one could sample gradients $\frac{\partial \mathcal{L}}{\partial a} \in \mathbb{R}^d$ where $\mathcal{L}$ is the loss function. Then $\left[\frac{\partial \mathcal{L}}{\partial a}\right]_i$ measures how much changing the $i$th neuron's output would affect the loss. This could be useful for explaining the types of examples where a pretrained model still has room for improvement. Similar to forward activations, its possible that PCs provide a significantly more interpretable summary than individual neurons since gradients are also low rank and high dimensional (Yang et al., 2024), (Zhao et al., 2024). The approach presented in our work could help to visualize and interpret dimensions of activation changes that would most affect the loss, but we leave this as future work.

## 6 Discussion and Conclusion

This study compares the standard neuron basis for explaining deep network function with the principal components analysis (PCA) basis, which is the simplest alternative. In our analysis of AlexNet, the PCA basis was both more complete and more interpretable. In particular, our user study showed that large-eigenvalue PCs represented more coherent stimulus transitions than neurons. Furthermore, most of the activation variance in a layer may be explained by dramatically fewer PCs than neurons, and ablating the highest-variance PCs usually affects performance more than ablating the highest-variance neurons (AlexNet's `conv1` layer is an exception). In contrast, small-eigenvalue PCs are uninterpretable, and they have a small effect on validation accuracy when ablated.

To explain deep networks in terms of neuron or PC activations is to expound rather than elucidate. This aim is closely related to the kind of mechanistic understanding that neuroscience has been seeking for decades. Neuroscience seeks to understand brain function at multiple levels, including how individual cells combine their inputs, how large networks transform information, and how this leads to behavior. Interestingly, it has been argued that neuroscience can benefit from high-level perspectives that deep learning takes for granted, such as the roles of architecture, loss, and data in network function (Tripp, 2018; Richards et al., 2019). At an intermediate level, much of neuroscience involves cataloguing how information is represented and transformed by particular groups of neurons in different parts of the brain, bridging the gap between cell function and behavior. As we discussed in Section 2, this work began with the study of individual neuron responses due to technical recording limitations, but population-level analyses (Kriegeskorte et al., 2008; Saxena & Cunningham, 2019; Ebitz & Hayden, 2021; Urai et al., 2022), including methods based on dimension reduction (Briggman et al., 2005; Shenoy et al., 2013; Aoi et al., 2020), are growing in importance, in parallel with recording density. Our results with PCs motivate future work in XAI with non-linear dimension reduction (Rombach et al., 2020; Carter et al., 2019; Templeton et al., 2024; Engels et al., 2024), which is also used in neuroscience (e.g. Cunningham & Yu, 2014; Niederhauser et al., 2022). More generally, this study supports the view that ANN explainability, like neuroscience, can benefit from a focus on population-level analysis.

**Broader Impact Statement**

While we believe our work can provide practically useful explanations for DNNs, all explainability methods carry the risk that users will be overconfident in the explanations. To help mitigate this risk, we highlight two limitations. Firstly, there are non-linear phenomena in activation space but the PCA basis only provides

a linear view of the underlying non-linear manifold (Gallego et al., 2017). For example, tuning curves in biological neural networks are captured by non-linear manifolds (Cammarata et al., 2020), (Gallego et al., 2017), (Georgopoulos et al., 1986), (Hubel & Wiesel, 1962), (Kriegeskorte & Wei, 2021), (Nover et al., 2005). Non-linear dimension reduction techniques can capture non-linear manifolds, unlike PCA. This is an important direction for future work because most stimuli are represented by multiple PCs.

Secondly, it is unclear what small eigenvalue PCs represent in activation space. Small eigenvalue PCs may be interpretable when considered in the context of their interaction with large eigenvalue PCs. They also may represent noise in activation space. Although they have uninterpretable visualizations and explain little variance, they still affect AlexNet's validation accuracy when ablated. Thus, small eigenvalue PCs play some role in network function. They may be needed for representing long-tail inputs that appear infrequently in the training set (Hooker et al., 2020a), (Hooker et al., 2020b). They could also be artifacts of the non-linear manifolds in activation space (Gallego et al., 2017), (Kriegeskorte & Wei, 2021).

### Acknowledgments

This research was supported by funding from BMO Bank of Montreal through the the Waterloo Artificial Intelligence Institute (SRA #081648). The authors thank Thomas Fortin for helping to run experiments with ResNet and ViT.

This research was supported, in part, by the Province of Ontario and the Government of Canada through the Canadian Institute for Advanced Research (CIFAR), and companies sponsoring the Vector Institute.

GWT is supported by the Natural Sciences and Engineering Research Council of Canada (NSERC), the Canada Research Chairs program, and the Canada CIFAR AI Chairs program.

This research was conducted with approval from the University of Guelph Research Ethics Board (REB #20-12-003).

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

## A  Explained variance of ResNet-18, ResNet-50, and ViT-B/16

In addition to the cumulative sum of explained variance results for AlexNet (Figure 5), we also include analogous plots for ImageNet-pretrained ResNet-18, ResNet-50, and ViT-B/16 in Figures 8, 9, and 10 respectively. ResNet-50 activations appear more strikingly low-rank due to the bottleneck residual block structure which constrains the activation rank to 1/4 of the full rank. For all networks studied, much of the activation variance is concentrated in the most important PCs whereas explained variance is far less concentrated in the neuron basis. Similar to AlexNet, these results suggest it is wasteful to omit dimensionality reduction when explaining ResNet-18, ResNet-50, or ViT-B/16.

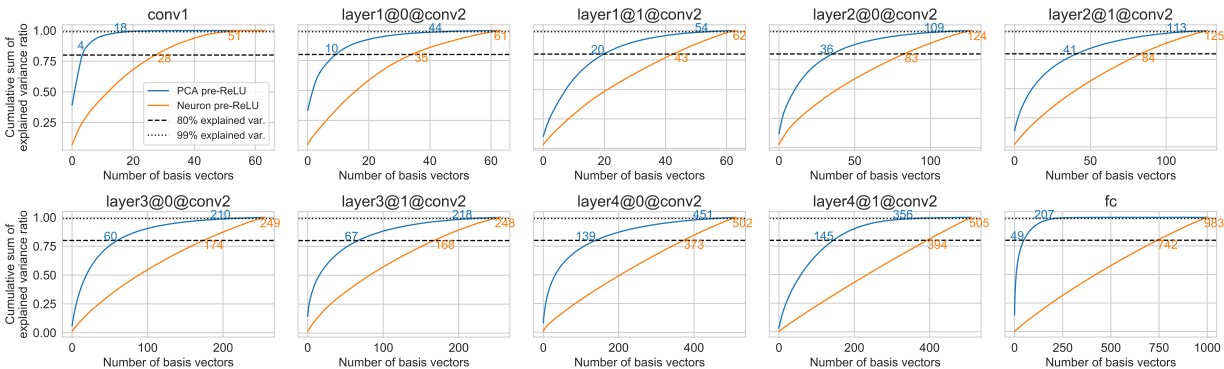

Figure 8: Cumulative sum of explained variance ratio for the final layer in each ResNet-18 residual block. Both PCs and neurons are ordered by descending variance. The number of basis vectors required to explain 80% and 99% variance is annotated.

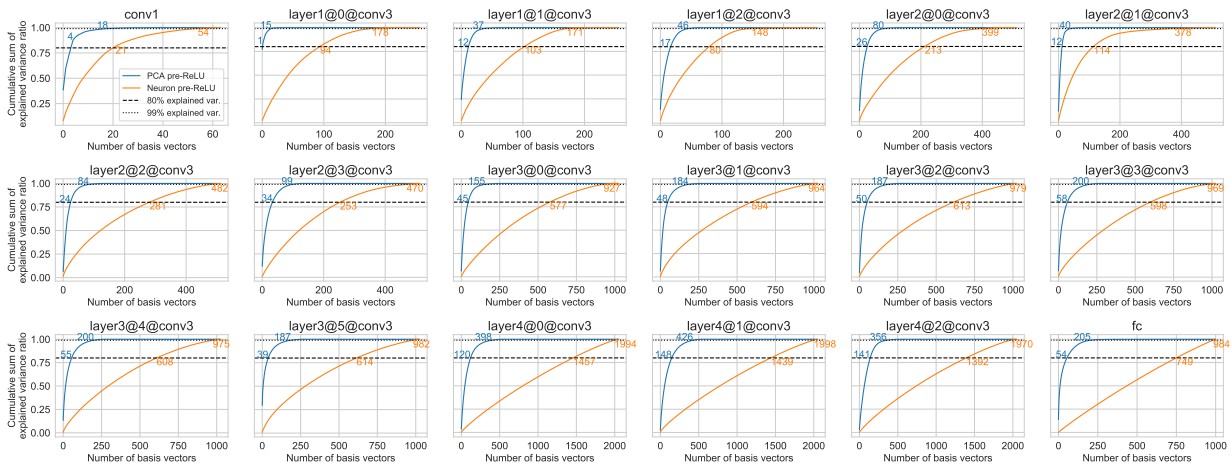

Figure 9: Cumulative sum of explained variance ratio for the final layer in each ResNet-50 residual block. Both PCs and neurons are ordered by descending variance. The number of basis vectors required to explain 80% and 99% variance is annotated.

## B Masking natural image regions that do not contribute strongly to activation similarity

The visualizations for deeper layers with larger receptive fields are more difficult to interpret than early layers with small receptive fields because there is more area in the natural image for distractors to be present. To help alleviate this problem, we mask out the regions of the visualizations that do not strongly affect the proximity of the representation to the point $\mathbf{a}$ in representation space that we are visualizing.

Let $h_I$, $w_I$, and $d_I$ be the height, width, and number of channels in each image patch $\mathbf{I}$. The sensitivity gradient $\mathbf{G} \in \mathbb{R}^{h_I \times w_I \times d_I}$ is the gradient of the $\ell_2$ distance between the target point $\mathbf{a}$ and $\mathbf{a}'$, the activation from forward propagating $\mathbf{I}$, with respect to the pixels in the image patch $\mathbf{I}$:

$$\mathbf{G} = \frac{\partial}{\partial I} \|\mathbf{a} - \mathbf{a}'\|. \tag{3}$$

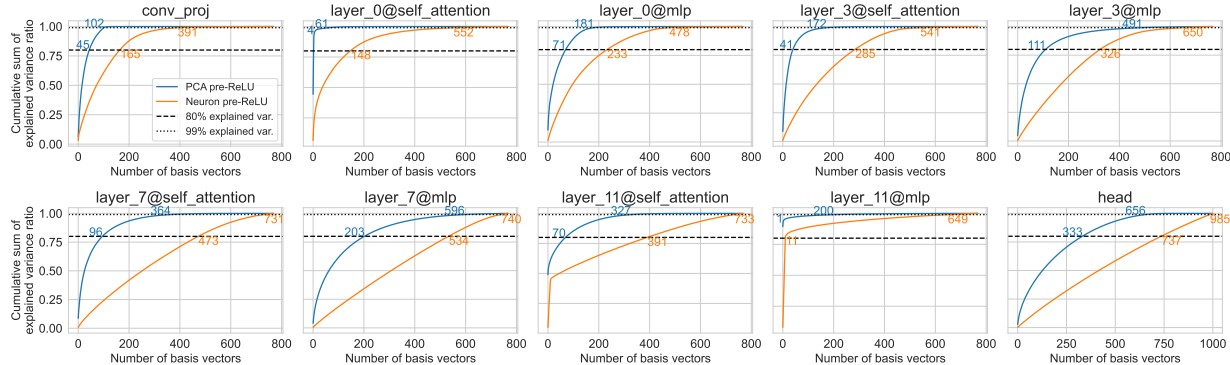

Figure 10: Cumulative sum of explained variance ratio for the ten layers in ViT-B/16. Both PCs and neurons are ordered by descending variance. The number of basis vectors required to explain 80% and 99% variance is annotated.

Next we take the sum of the absolute gradient values over channels and apply a 2D Gaussian filter $g(x, y, \sigma)$, where $\sigma = 3$, to obtain $G_{\text{smooth}} \in \mathbb{R}^{h_I \times w_I}$ as follows:

$$\mathbf{G}_{\text{smooth}} = g(x, y, \sigma) * (|\sum_i^{d_I} (\mathbf{G}_i)|). \tag{4}$$

To determine the Boolean image patch mask $\mathbf{M} \in \mathbb{R}^{h_I \times w_I}$, we threshold $\mathbf{G}_{\text{smooth}}$ to be greater than one standard deviation of $\mathbf{G}_{\text{smooth}}$ as follows:

$$\mathbf{M} = \mathbf{G}_{\text{smooth}} > \text{std}(\mathbf{G}_{\text{smooth}}). \tag{5}$$

The resulting mask $\mathbf{M}$ is applied element-wise to the image patch as $\mathbf{I} \odot \mathbf{M}$ to hide regions of the image that do not strongly affect the distance $\|\mathbf{a} - \mathbf{a}'\|$.

## C    Comparison of methods for sampling points along basis vectors

We sample and visualize $m$ points along a basis vector with the goal of understanding how activation variance along the basis vector relates to input variance. We compared two strategies: sampling between the first and ninety-ninth percentile, or sampling between the minimum and maximum values along the basis vector. After studying several basis vector visualizations from each AlexNet layer, we concluded that sampling between the minimum and maximum values produced more interpretable visualizations.

In Figure 11 we show a comparison of the two strategies for the second PC of `conv1`, which represents the Gabor phase of a vertical edge. Although the first and ninety-ninth percentile visualizations demonstrate the concept of Gabor phase, the extreme values along each basis vector produce visualizations that demonstrate the concept more distinctly.

## D    Activation maximization versus distance minimization

When we visualize basis vectors, we use **distance minimization** rather than the more common **activation maximization** objective. **Activation maximization** involves producing a visualization that maximizes the activation along some basis vector (Mahendran & Vedaldi, 2015), (Nguyen et al., 2016a), (Nguyen et al., 2016b), (Olah et al., 2017), (Olah et al., 2018). One issue with activation maximization is that there are many degrees of freedom; to maximize neuron 1, a visualization may also elicit large activations from neuron 2 and 3. Geirhos et al. (2023) show how this freedom makes activation maximization visualizations unreliable. It is difficult to precisely study distributed representations because neurons cannot be studied independently

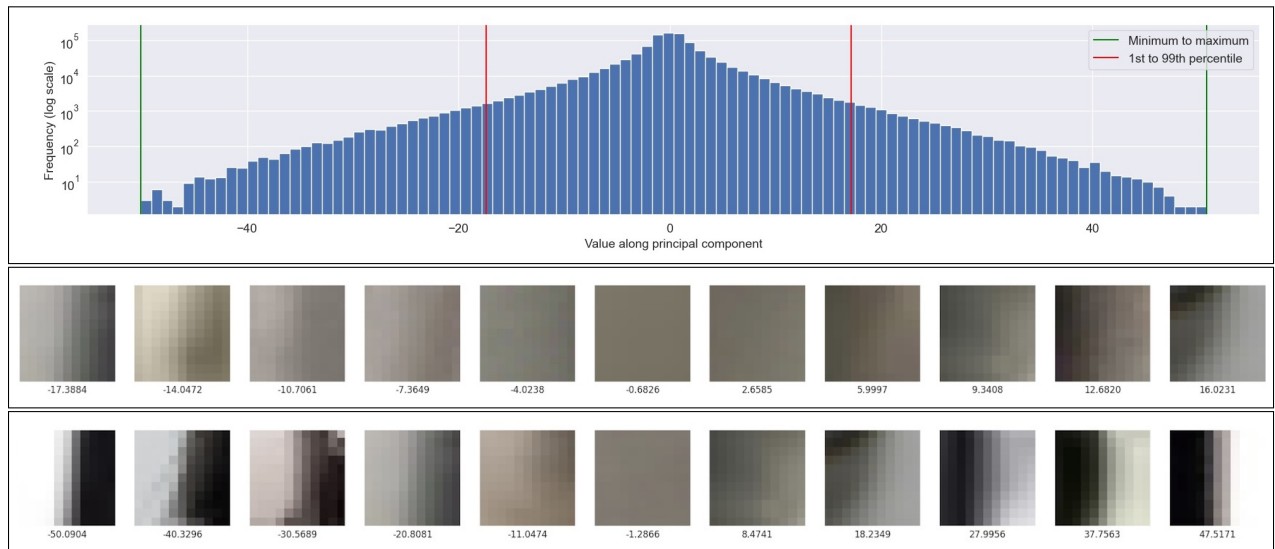

Figure 11: **Top:** Histogram of activations from the ImageNet training set, projected onto the second PC (PC 2) of `conv1` (log scale). The interval defined by the minimum and maximum is shown in green. The interval defined by the first and ninety-ninth percentile is shown in red. **Middle and bottom:** Visualizations of points along PC 2 of `conv1`, uniformly sampled between the first and ninety-ninth percentile (**Middle**), and uniformly sampled between the minimum and maximum (**Bottom**).

of each other. A second issue is that we wished to characterize how representations change along a given dimension, whereas activation maximization addresses only the extremes. **Distance minimization** involves producing a visualization that minimizes the distance between the visualization's response in activation space, and some target point in activation space. This allows us to study how variance along a single basis vector in activation space relates to variance in input space. The difference between these two objectives is illustrated in Figure 12.

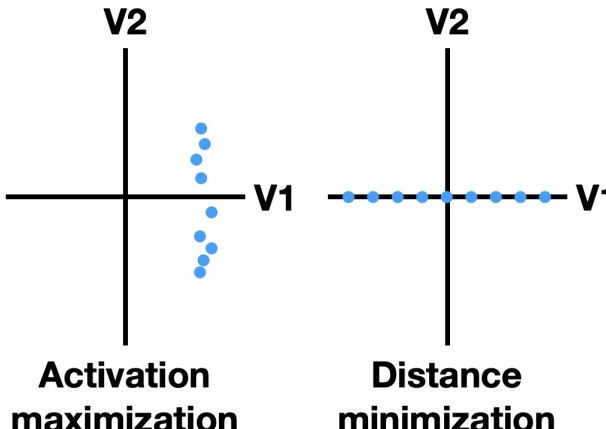

Figure 12: Toy example illustrating the difference between activation maximization and distance minimization when attempting to visualize the basis vector V1. Blue dots represent the resulting optimized visualizations.

## E    Computational resources

Our computations were performed on machines with an 8 core Intel Xeon CPU, NVIDIA T4 GPU, and 64 GB of RAM. Sampling activations from an AlexNet layer for the ImageNet training set ($n = \sim$1M) took approximately 2 hours. The runtime of fitting PCA to the sampled activations $\mathbf{A}$ is dependent on the dimensions of $\mathbf{A}$, and can take anywhere from 11 seconds for `conv1` ($d = 64$) to 22 minutes for `fc1` ($d = 4096$). Producing natural visualizations of 32 points along a basis vector, with 5 neighbors, takes approximately 4 seconds.

## F    Additional User Study Details

We recruited 22 participants from Prolific [3] in exchange for £6.6 GBP/hr. All participants provided informed consent to use their anonymized data in accordance with the institutional review board.

The scrambled re-ordering for comparison images followed some constraints. When piloting the task with fully random scrambling, we discovered it was trivially easy if the NN row contained duplicate snippets, because they would not appear next to each other in the scrambled version (detecting two or more identical snippets not adjacent was a giveaway that it was the scrambled stimulus, and could be used as a signal to complete the task). To remedy this, comparison images were pseudorandomly ordered such that NN snippets that appeared multiple times within a row were required to be horizontally adjacent.

The full text instructions presented to each participant in the user study are shown in Fig. 13. Participants read these instructions at their own pace and proceeded with a button press. Fig. 14 shows a screenshot of the user study task screen.The distributions of correct responses for each AlexNet layer are shown in Figure 15.

## G    Additional visualizations

To view visualizations PC and neuron visualizations for every layer of pre-trained AlexNet (Krizhevsky, 2014), in addition to PC visualizations from every block of pre-trained ResNet-18 and ResNet-50 (He et al., 2016) and 10 layers of ViT-B/16 (Dosovitskiy et al., 2021), please refer to our **interactive demo**: `https://ndey96.github.io/neuron-explanations-sacrifice`.

It is difficult to exhaustively show our visualizations in a paper so we have chosen to show the visualizations of the 5 highest variance PCs and neurons for each AlexNet layer in Figures 2, 3, 16, 17, 18, 4, 19, and 20. After `conv1`, the neuron basis becomes quite uninterpretable, as confirmed through our user study.

---

[3]https://www.prolific.co

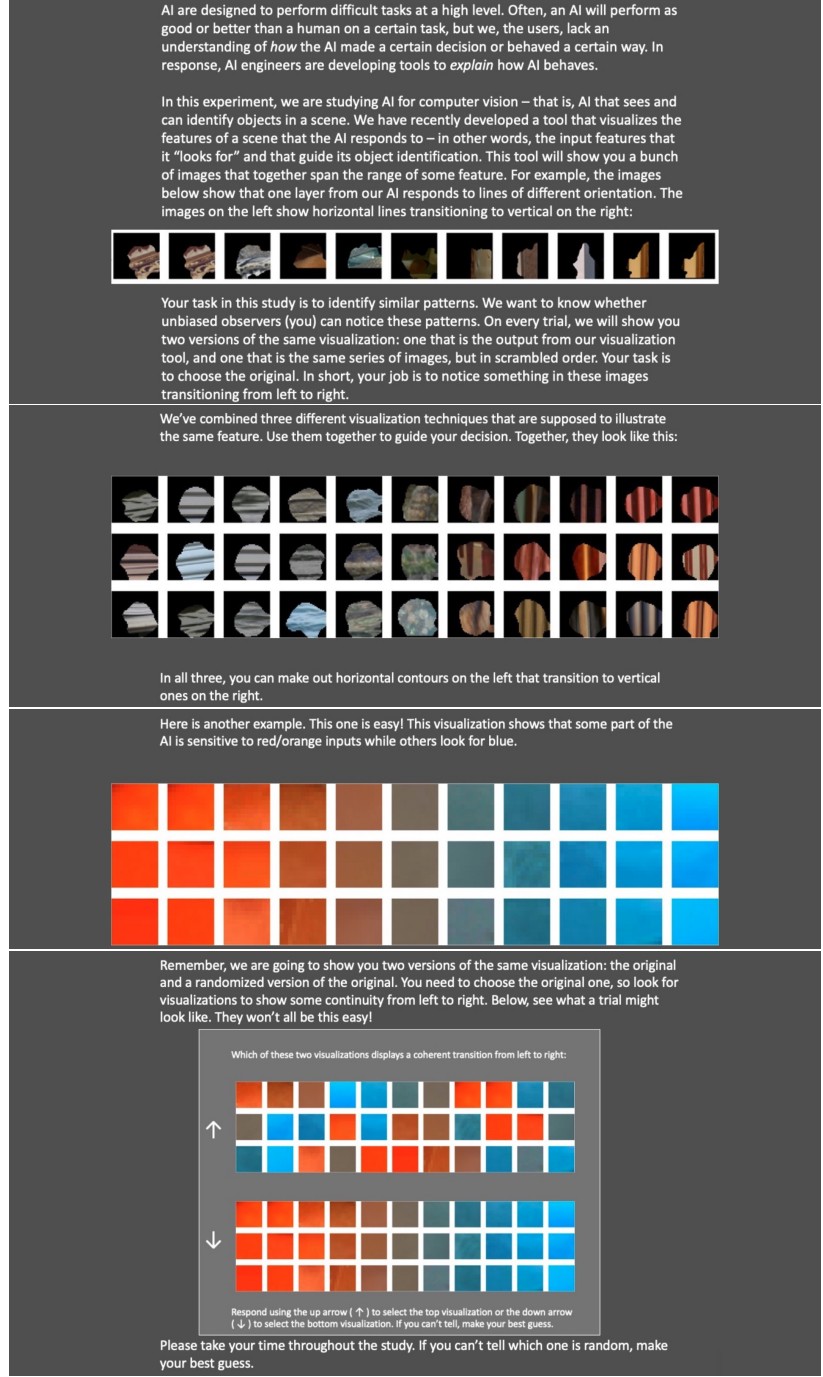

Figure 13: Full text instructions presented to participants of the user study.

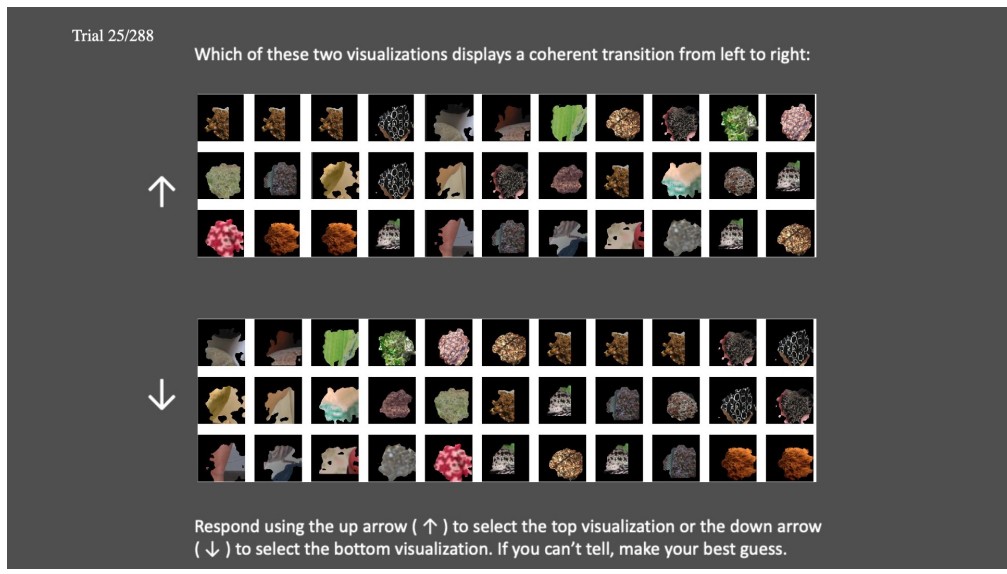

Figure 14: Screenshot of the user study task.

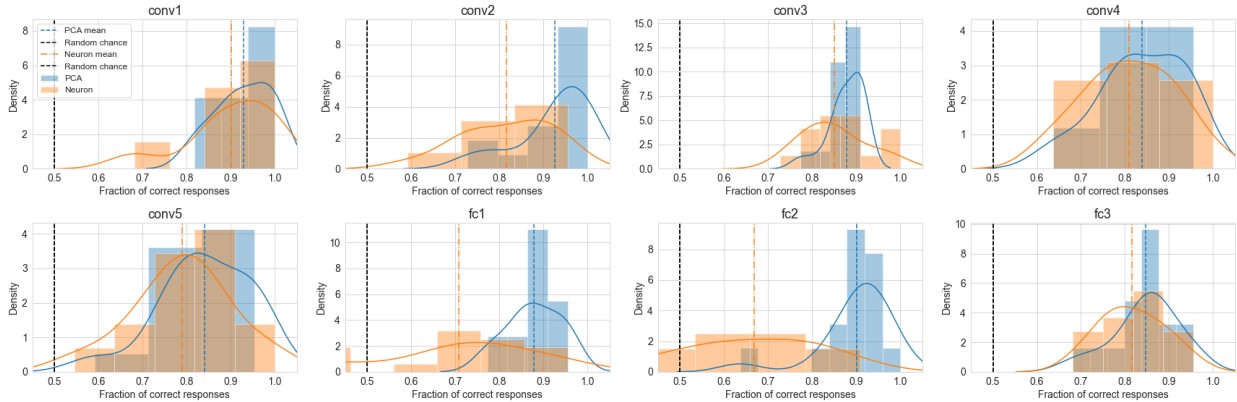

Figure 15: Distribution of correct responses for PCA and Neuron visualizations, plotted for each AlexNet layer.

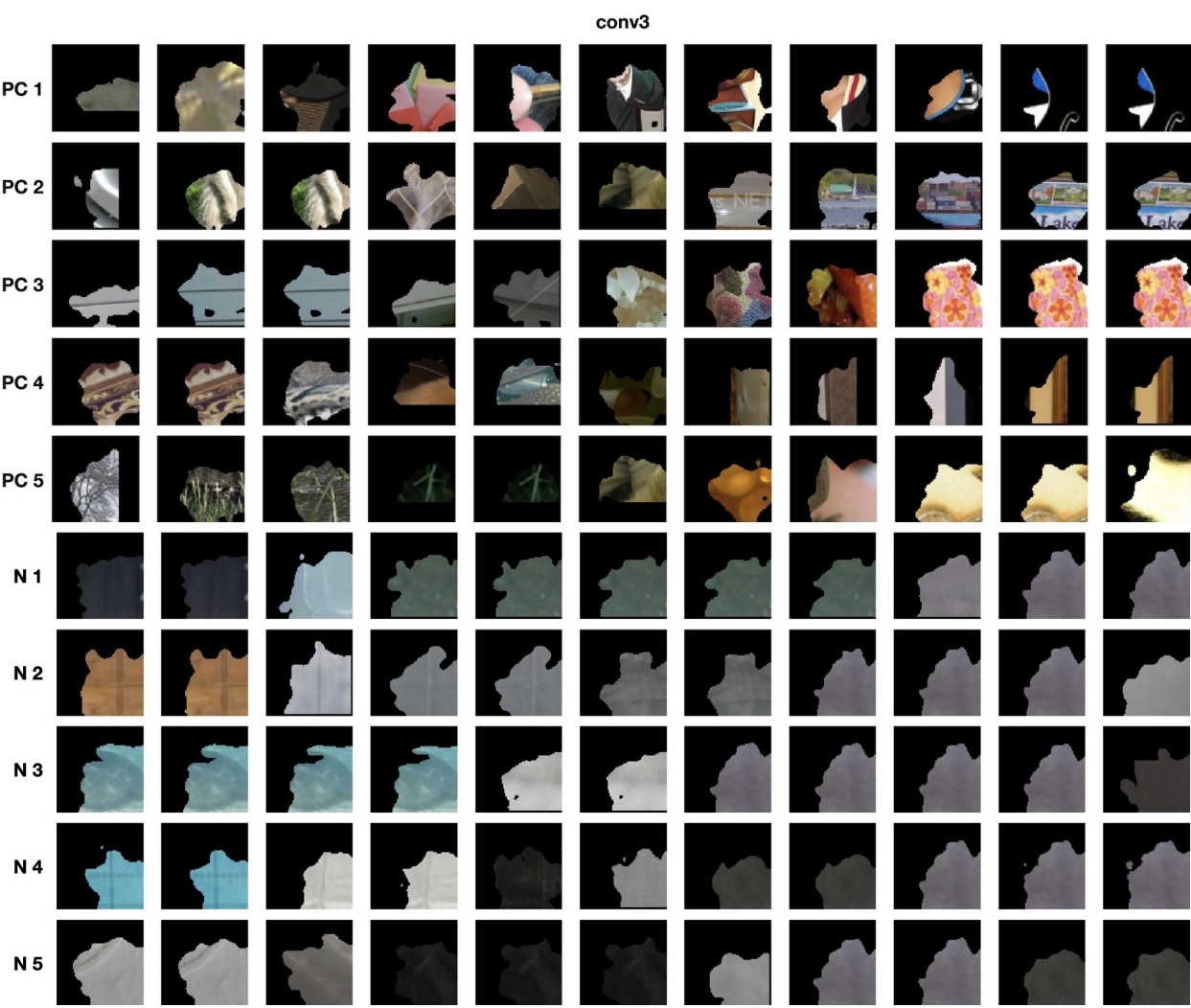

Figure 16: Visualizations of points along the 5 highest variance PCs and 5 highest variance neurons for AlexNet's `conv3` layer.

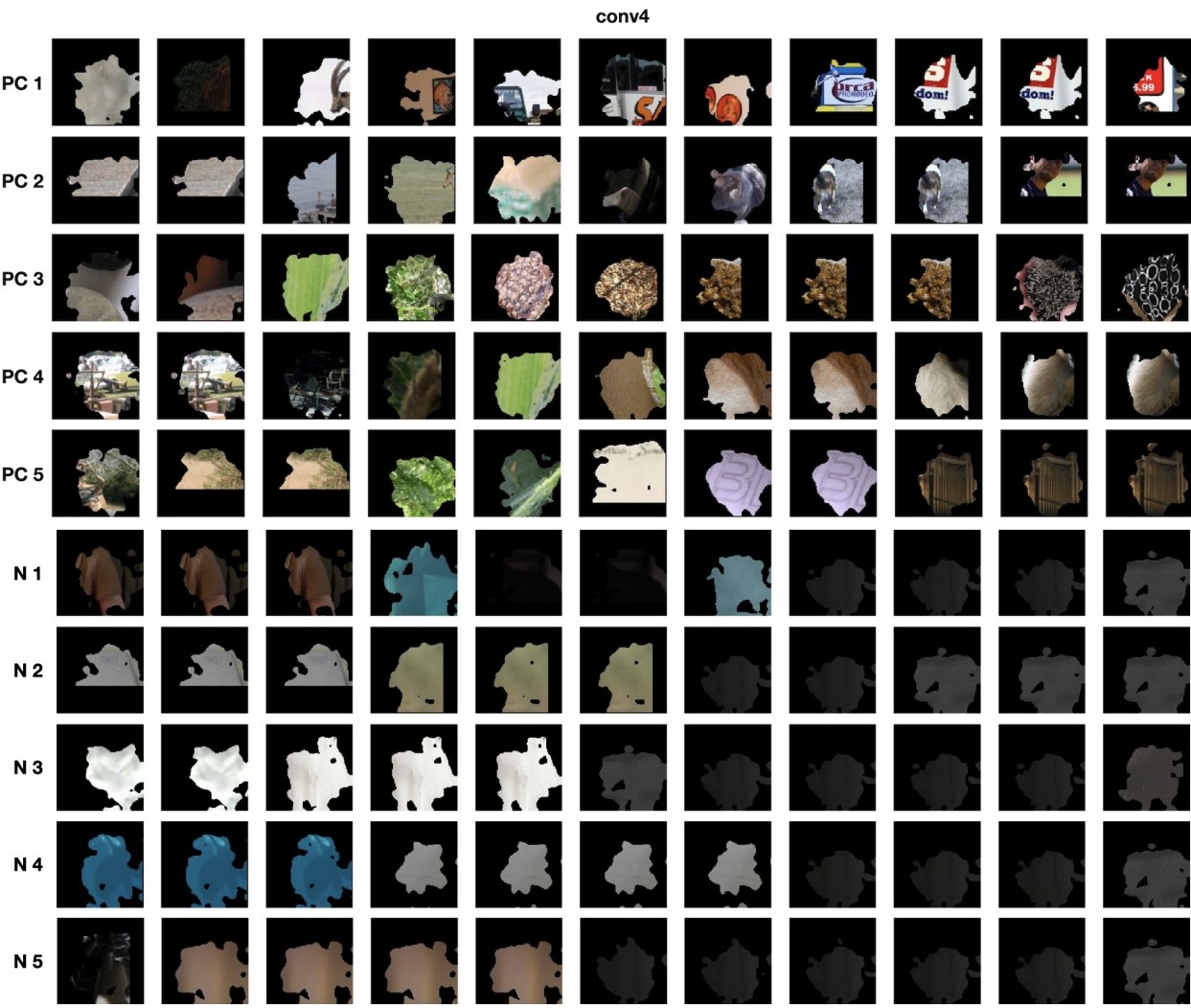

Figure 17: Visualizations of points along the 5 highest variance PCs and 5 highest variance neurons for AlexNet's `conv4` layer.

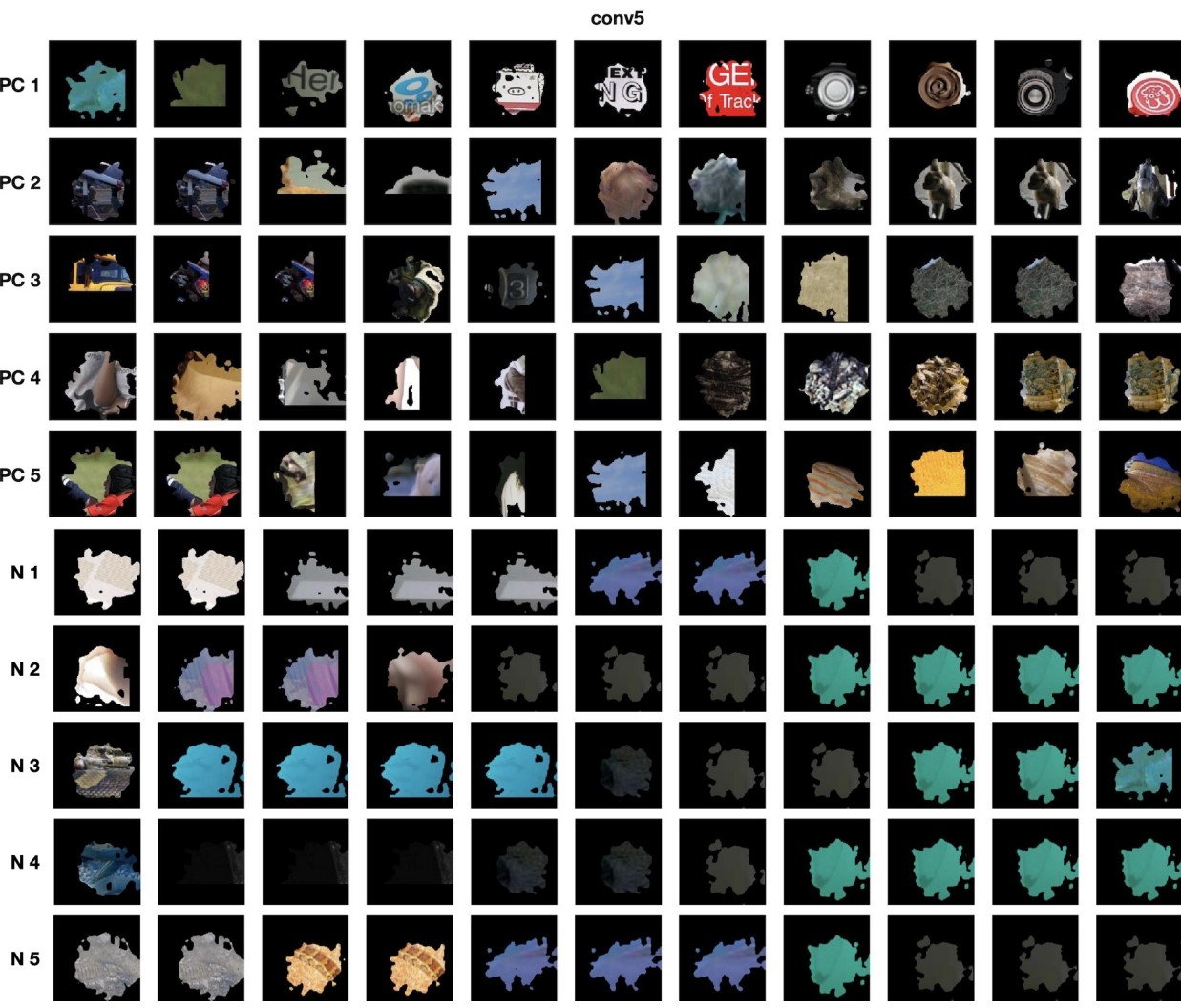

Figure 18: Visualizations of points along the 5 highest variance PCs and 5 highest variance neurons for AlexNet's conv5 layer.

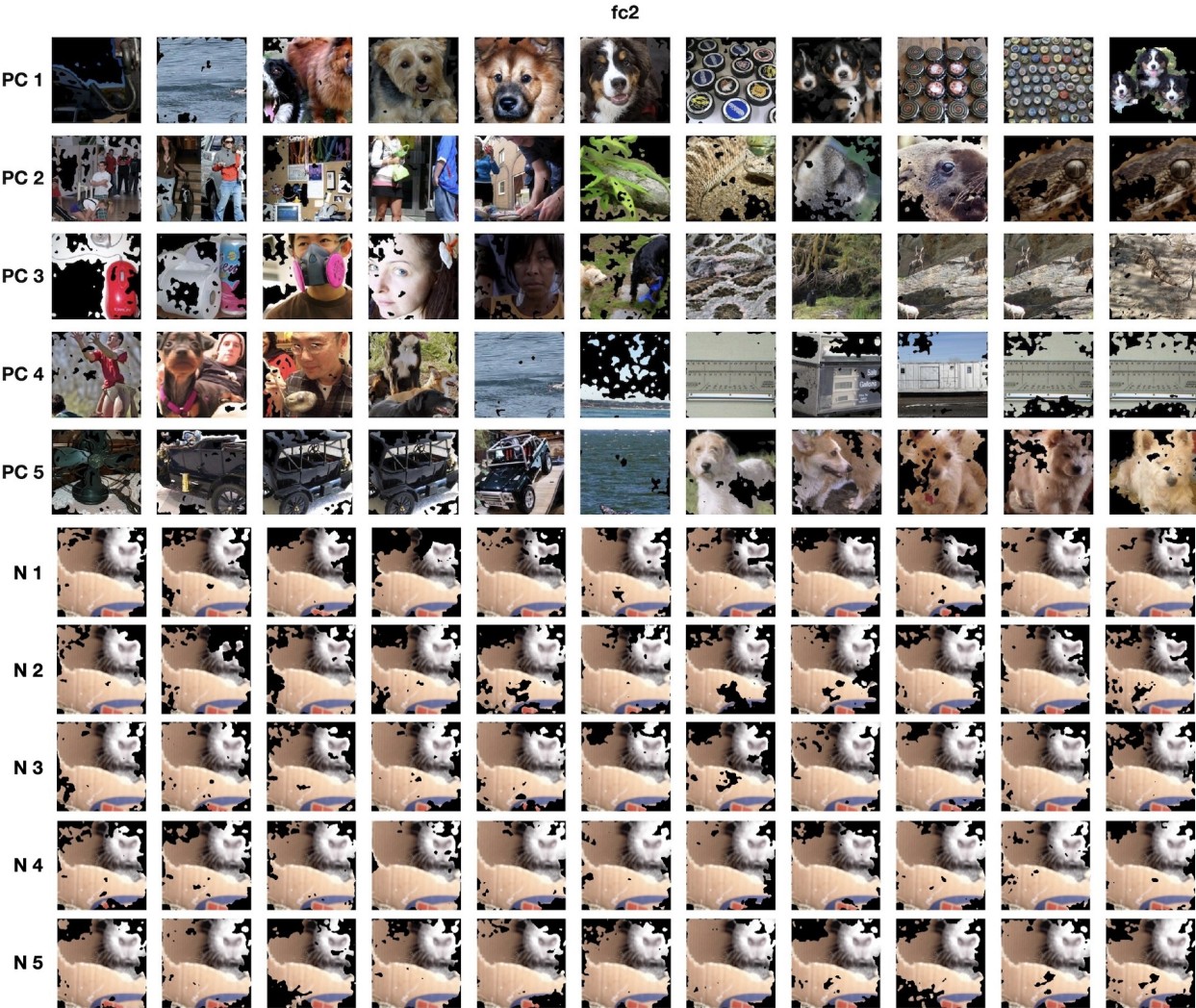

Figure 19: Visualizations of points along the 5 highest variance PCs and 5 highest variance neurons for AlexNet's `fc2` layer.

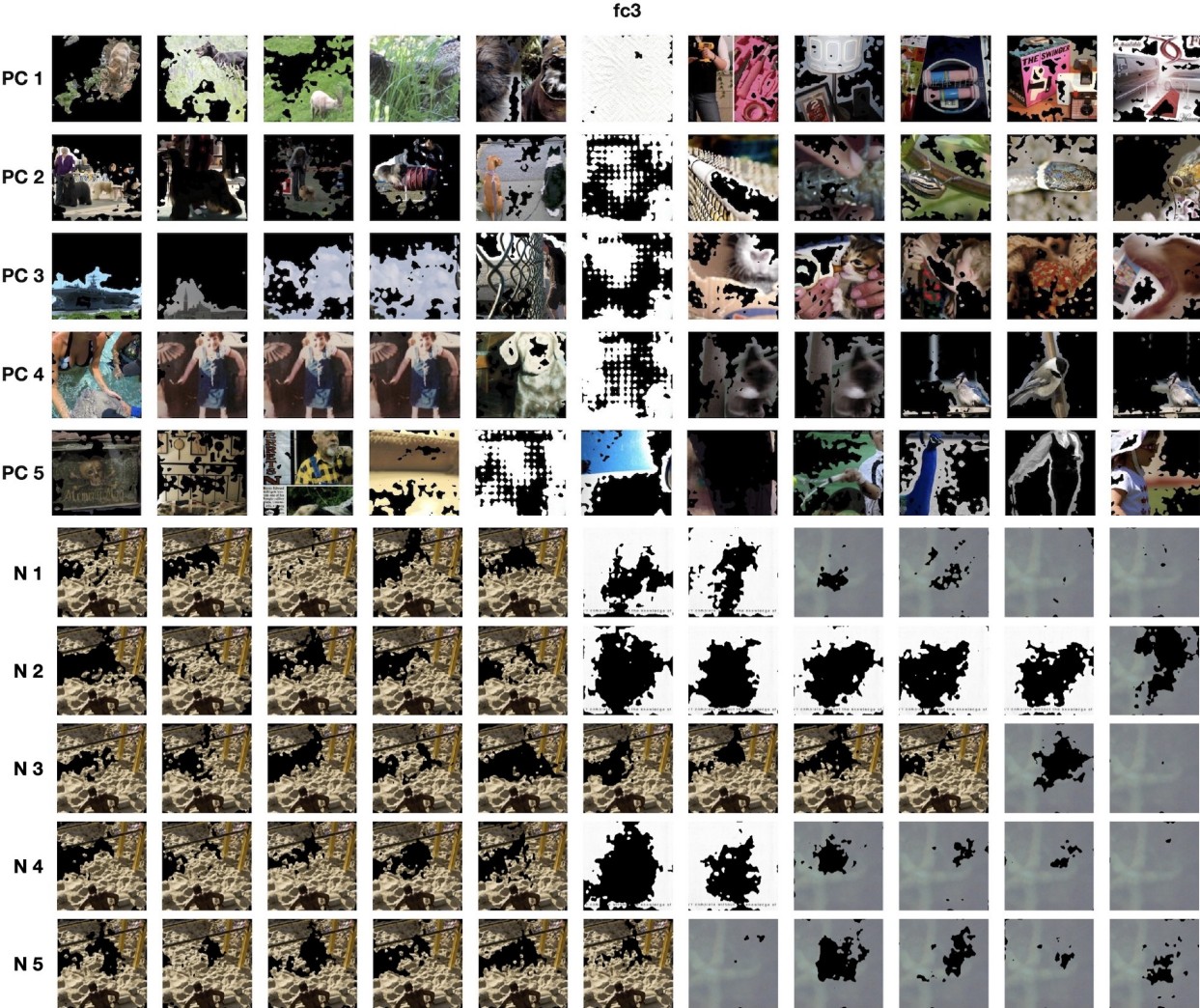

Figure 20: Visualizations of points along the 5 highest variance PCs and 5 highest variance neurons for AlexNet's `fc3` layer.

