# OpenReview forum: "Neuron-based explanations of neural networks sacrifice completeness and interpretability"
_TMLR — Accepted by TMLR_

### Review · Reviewer_matW · 2024-08-11

**Summary Of Contributions:**

In the submission, the authors argue that neurons are a poor basis for the embeddings of deep neural networks (DNNs) and propose a new method for explaining the intermediate representations of DNNs based on dimensionality reduction, or more specifically, principal component analysis (PCA). For each layer in DNNs, we first perform PCA on samples drawn from the activations obtained by performing forward propagation. Then we find the orthogonal basis vectors and sample points along each basis vector. Finally, these points are visualized in 2D using t-SNE and interpreted in plain text. The authors test this methodology on AlexNet pre-trained on ImageNet and evaluate both the completeness and interpretability of the explanations using PCs and neurons. The results indicate that the most important PCs provide more complete explanations than the most important neurons, and the visualizations based on PCs are also more interpretable than those based on neurons.

**Audience:**

Yes

**Broader Impact Concerns:**

I do not foresee any ethical concerns in this work, and the authors have appropriately included a Broader Impact statement in the submission.

**Claims And Evidence:**

Yes

**Requested Changes:**

•	The comparison of visualizations from PCA and individual neurons in Appendix F provides a good demonstration of the interpretability and it would be better to move some part of the results into the experiments in the main body.

•	In Figure 5, it appears that for some **intermediate** convolutional layers (e.g., $\texttt{conv4}$, $\texttt{conv5}$), the PCs exhibit a similar rate of performance degradation compared to neurons as ablation continues. This behavior seems to largely differ from that observed in fully connected layers. Could this phenomenon be related to the method used for sampling spatial positions in the convolutional layers? A more detailed explanation would be appreciated.

**Strengths And Weaknesses:**

Strengths:

•	Explainable deep learning is a critical area of research that holds significant potential for advancing the application of deep learning across various scientific domains. The topic is expected to be of considerable interest to the general audience of TMLR.

•	The presentation of methodology is clear and accessible, and the illustration in Figure 1 is very helpful in understanding the big picture of methodology.

•	The authors provide a clear and interactive demo that contains feature visualizations of neurons and PCs.

•	The experiments include a user study to show the explainability of PC visualizations and the hypotheses are evaluated by conducting statistical analysis.

Weaknesses:

It appears that the explanations provided for PCs and neurons in this paper are based solely on the representations generated during the forward pass. However, in my opinion, understanding the backward pass of DNNs is equally important for grasping the features learned by each layer during training. It would be beneficial to show or at least discuss how the proposed procedure could be adapted to analyze the backward pass. For instance, as noted in Section 4.1.1, PCA-based explanations require high inter-neuron covariance. Does this requirement consistently hold throughout the training process of DNNs?

---

> ### Author Response · Authors · 2024-10-21
> **Rebuttal**
>
> Thank you very much for your thoughtful review. It is gratifying to know you found the presentation clear and accessible, and consider the experimental results a strength of the paper.
>
> ```
> It appears that the explanations provided for PCs and neurons in this paper are based solely on the representations generated during the forward pass. However, in my opinion, understanding the backward pass of DNNs is equally important for grasping the features learned by each layer during training. It would be beneficial to show or at least discuss how the proposed procedure could be adapted to analyze the backward pass. For instance, as noted in Section 4.1.1, PCA-based explanations require high inter-neuron covariance. Does this requirement consistently hold throughout the training process of DNNs?
> ```
> This is very perceptive -- and a direction that we have not considered previously.
>
> Instead of sampling forward activations $a \in \mathbb{R}^d$, one could sample gradients $\frac{\partial \mathcal{L}}{\partial a} \in \mathbb{R}^d$ where $\mathcal{L}$ is the loss function. Then $\left[ \frac{\partial \mathcal{L}}{\partial a} \right]_i$ measures how much changing the $i$th neuron's output would affect the loss. This could be useful for explaining the types of examples where a pretrained model still has room for improvement. Similar to forward activations, its possible that PCs provide a significantly more interpretable summary than individual neurons since gradients are also low rank and high dimensional [1,2]. The approach presented in our work could help to visualize and interpret dimensions of activation changes that would most affect the loss.
>
> Given focus of this paper is to critique the practice of relying too heavily on neuron-based explanations, we think this makes a nice addition to the limitations section of the paper and motivates future work. Please see our revised submission.
>
> [1] Greg Yang, James B. Simon, Jeremy Bernstein. ``A Spectral Condition for Feature Learning'', 2023
>
> [2] Jiawei Zhao, Zhenyu Zhang, Beidi Chen, Zhangyang Wang, Anima Anandkumar, Yuandong Tian. ``GaLore: Memory-Efficient LLM Training by Gradient Low-Rank Projection'', 2024
>
> ```
> The comparison of visualizations from PCA and individual neurons in Appendix F provides a good demonstration of the interpretability and it would be better to move some part of the results into the experiments in the main body.
> ```
> Excellent suggestion. We were struggling with how much content from the demo to include in the main body since the figures can take up a lot of space. In our revised submission, we moved the neuron visualizations for conv1, conv2, and fc1 to the main body so it is easier to compare these to the PC visualizations within the main body. We also revised the neuron visualizations to be the top 5 highest variance neurons rather than the first 6 neurons by index. Throughout the appendix we also modified the format to pair the PC and neuron visualizations for each layer. Overall we think this has significantly improved the presentation so thank you!
> ```
> In Figure 5, it appears that for some intermediate convolutional layers (e.g., conv4, conv5), the PCs exhibit a similar rate of performance degradation compared to neurons as ablation continues. This behavior seems to largely differ from that observed in fully connected layers. Could this phenomenon be related to the method used for sampling spatial positions in the convolutional layers? A more detailed explanation would be appreciated.
> ```
> We also found this surprising. We sampled 1M image patches which is likely more than enough to adequately capture variance along neurons or PCs. Instead we have an alternate explanation. In our revised submission, we annotate the number basis vectors required to explain 99% of explained variance in Figure 4. Notably, conv4 and conv5 require a similar number of PCs and neurons to explain 99% of the activation variance, implying these activations have a higher rank structure and PCA is a less suitable decomposition method compared to other layers. In other words, conv4 and conv5 have heavier tailed activation eigenspectra compared to the other layers. We hypothesize this explains why ablating the ~50 highest variance PCs degrades accuracy faster than neurons, but after that point it is more damaging to ablate neurons. In our revised submission we have updated Section 4.1.1 and Section 4.1.2 with these comments. Thanks for pointing this out!

---

### Review · Reviewer_HkFH · 2024-09-02

**Summary Of Contributions:**

In this paper, the authors demonstrate that the principal components of activation provide a more useful method for DNN analysis compared to neuron-based methods. They quantitatively measure completeness (using % of variance captured, and activation ablation) and interpretability (using user studies) to demonstrate this, and conclude that using activation based explanation methods is better.

**Audience:**

Yes

**Broader Impact Concerns:**

None.

**Claims And Evidence:**

Yes

**Requested Changes:**

Would be useful to see a detailed and honest discussion of the limitations of the approach here, and perhaps provide some counter-examples (constructed cases where neuron-based explanations might be better).

**Strengths And Weaknesses:**

The study is very well written, clear and to the point. The motivation is well explained, and the experiments well justified. The paper very focused and demonstrates, using the chosen measures, that activation based explainability is superior.

While the overall conclusion is not entirely novel, having been hinted at in various places before, this study collects clear evidence in support of activation-based explanations.

The interactive demo is also very nicely done.

While I'm convinced of the conclusion of the paper, perhaps a weakness of this paper is that it doesn't have a honest discussion of the limitations of their approach.

---

> ### Author Response · Authors · 2024-10-21
> **Rebuttal**
>
> Thank you very much for the helpful comments, and for your appreciation of the presentation, experiments, and interactive demo.
> ```
> Would be useful to see a detailed and honest discussion of the limitations of the approach here, and perhaps provide some counter-examples (constructed cases where neuron-based explanations might be better).
> ```
> This is a great suggestion! Neuron-based explanations should be most useful in the first and final layer of a network. We revised our submission with a Limitations section that includes a paragraph on this topic.

---

### Review · Reviewer_EkJK · 2024-10-07

**Summary Of Contributions:**

This paper argues that neuron-based explanations on deep neural networks are insufficient. The paper claims that this approach fails to accurately represent a network's function and makes the interpretation difficult to understand for humans. The paper proposes to instead utilize the activation principal components for neural network explanations.

**Audience:**

Yes

**Claims And Evidence:**

No

**Requested Changes:**

- More experiments on other datasets / deep learning architectures.
- A discussion on how these insights are different from the traditional understanding of PCA.
- More details on how the visualization technique as well as Figure 4.

**Strengths And Weaknesses:**

## Strengths
- Understanding the inner workings of deep neural networks is an important problem.
- The website demo can be a very useful tool for explainibility analysis of deep neural networks.

## Weaknesses
Overall while the authors are studying an important problem the paper is quite unclear about a lot of key details. Moreover, the empirical evidence is severly lacking.

- The paper cites [1] for how they're visualizing the points along PCs. Some background on how this visualization technique works would be helpful.

- I am also a bit confused about Fig. 4. It seems to be confirming a well known fact which is that the top PCs of a subspace account for the majority of the variance. However, it is not clear to me what the authors mean by "explained variance" or "activation variance". Could you provide a mathematical expression for what this quantity is?

- The paper only studies AlexNet trained on ImageNet. Given that this paper is trying to make a general claim about the advantages of PC vs. neuron basis the authors should include experiments on other deep learning architectures.

[1] Andrej Karpathy. t-sne visualization of cnn codes. https://cs.stanford.edu/people/karpathy/ cnnembed/, 2014.

---

> ### Author Response · Authors · 2024-10-21
> **Rebuttal**
>
> Thank you for your useful suggestions and your positive comments regarding the potential impact of the website demo for explainability analysis.
>
> ```
> The paper cites [1] for how they're visualizing the points along PCs. Some background on how this visualization technique works would be helpful.
> [1] Andrej Karpathy. t-sne visualization of cnn codes., 2014.
>
> ... More details on how the visualization technique
> ```
> We are happy to provide a summary of the visualization technique in the revision. We have updated the relevant text in Section 3.3 as follows: Karpathy (2014) visualized a point in activation space $\mathbf{a}\_{\text{target}} \in \mathbb{R}^d$ by displaying the receptive-field sized image patch whose activations have with the lowest $\ell_2$ distance to the point $\mathbf{a}\_{\text{target}}$. In this work we extend Karpathy's method by displaying the $k$ nearest receptive-field sized image patches that have the lowest $\ell_2$ distance to $\mathbf{a}\_{\text{target}}$ when forward propagated.
>
> ```
> I am also a bit confused about Fig. 4. It seems to be confirming a well known fact which is that the top PCs of a subspace account for the majority of the variance. However, it is not clear to me what the authors mean by "explained variance" or "activation variance". Could you provide a mathematical expression for what this quantity is?
>
> ... A discussion on how these insights are different from the traditional understanding of PCA.
> ```
> Thank you for pointing out this lack of clarity. We have updated Section 4.1.1 with these details and edited the y-axis labels of Figure 4 to be more descriptive. "Activation variance" is also used repeatedly but not explained until page 8. We also updated Section 3.2 with a brief definition of activation variance to improve clarity: "PCA finds orthogonal basis vectors for $\mathbb{R}^d$ activation space, ordered by explained variance ($\Var(\mathbf{A}'_i)$ for the $i$th basis vector)"
>
> Regarding the fact that top PCs account for the majority of variance, our goal with Figure 4 was simply to show the degree of this concentration for the AlexNet layers, because in general a given fraction of variance could be present in a smaller or greater number of top PCs. There is no difference from the traditional understanding of PCA in this analysis (whereas the ablation and user-study analysis make qualitatively different points).
> ```
> The paper only studies AlexNet trained on ImageNet. Given that this paper is trying to make a general claim about the advantages of PC vs. neuron basis the authors should include experiments on other deep learning architectures.
> ```
> Thank you for pointing out the issues with the generality of our claim. We focused on AlexNet because it is one of the most commonly studied network in the explainability literature, as discussed early in Section 4. Nevertheless, we agree that this is a fair criticism. We will take three actions to mitigate this weakness. First, although it makes sense to introduce the problem in general terms, we will be careful to avoid making claims that are more general than AlexNet. We have updated our abstract to limit our claims to AlexNet. Second, although we are unable to run formal user studies with additional networks, we added cumulative sum of explained variance ratio plots for ImageNet-pretrained ResNet-18 and ResNet-50 in Appendix F. Similar to AlexNet, the activation variance is much more concentrated in the high variance PCs compared to the high variance neurons, suggesting our findings may generalize to these networks. In our online demo, we include visualizations of the top 16 PCs for several layers in ResNet-18 and ResNet-50. Third, we have updated our manuscript with a limitations section to explicitly state the potential lack of generality that comes with only studying AlexNet deeply.

---

> > ### Author Response · Authors · 2024-11-01
> > **Expanding online demo**
> >
> > To further address the issue of generality we are in the process of expanding the online demo. We will add ResNet neuron visualizations at minimum so that they can be compared with the existing ResNet PC visualizations. We will post another update on this point by Nov. 4.

---

> > > ### Author Response · Authors · 2024-11-04
> > > **Online demo status**
> > >
> > > To expand the online demo, we have collected visualizations of top neurons for ResNet-18 (to compare with the PC visualizations that are already part of the demo) and both neuron and PC visualizations for a vision transformer. These seem to show patterns consistent with our results from AlexNet. We will post these online as soon as possible.

---

> > > > ### Author Response · Authors · 2024-11-05
> > > > **Online demo updated with ResNet-18 neurons, and ViT-B-16 PCs and neurons**
> > > >
> > > > The online demo (https://david-s-hippocampus.github.io/tmlr_demo/) is now updated with:
> > > > 1. Visualizations of top neurons for ResNet-18 (to compare with the PC visualizations that are already part of the demo)
> > > > 2. Both top neurons and PC visualizations for ViT-B-16
> > > > 3. All neurons are presented in descending order of variance rather than their actual index. This allows a more fair comparison between neuron and PC interpretability within the demo
> > > >
> > > > These seem to show patterns consistent with our results from AlexNet - neuron based explanations sacrifice completeness (appendix F) and interpretability (online demo). Additionally we can update the manuscript to reflect these new results.
> > > >
> > > > Thank you

---

### Decision · Action_Editor_7jME · 2025-03-01

**Recommendation:** Accept as is

**Comment:**

The authors have presented clear evidence in experiments to show that the most important principal components provide more complete and interpretable explanations than the most important neurons. Major concerns of reviewers have been addressed in the rebuttal.

**Audience:**

All researchers in the community of deep learning are potential audience of this paper.

**Claims And Evidence:**

The authors have presented clear evidence in experiments to show that the most important principal components provide more complete and interpretable explanations than the most important neurons. Major concerns of reviewers have been addressed in the rebuttal.